# Variational Masked Diffusion Models

## Abstract

Masked diffusion models have recently emerged as a flexible framework for discrete generative modeling. However, a key limitation of standard masked diffusion is its inability to effectively capture dependencies among tokens that are predicted concurrently, leading to degraded generation quality when dependencies among tokens are important. To explicitly model dependencies among tokens, we propose Variational Masked Diffusion (VMD), a framework that introduces latent variables into the masked diffusion process. Through controlled experiments on synthetic datasets, we demonstrate that VMD successfully learns dependencies that conventional masked diffusion fails to capture. We further validate the effectiveness of our approach on Sudoku puzzles and text datasets, where learning of dependencies among tokens improves global consistency. Across these domains, VMD enhances both generation quality and dependency awareness, highlighting the value of integrating variational inference into masked diffusion.

## 1 Introduction

Diffusion-based large language models (DLLMs) represent a significant architectural innovation, emerging as a compelling extension of autoregressive models (ARMs). This paradigm shift is driven by the inherent limitations of traditional ARMs, which generate tokens sequentially and in a pre-defined order. In contrast, DLLMs offer distinct advantages, including concurrent token generation, superior output diversity, enhanced global coherence, advanced controllability over the generated text, and presumably easier integration of heterogeneous data, where a fixed pre-defined sequential token order is harder to justify. Recent breakthroughs, exemplified by models like LLaDA (Nie et al., 2025), Mercury (Inception Labs, 2025), and Gemini Diffusion (Google DeepMind, 2025), underscore the increasing viability and promising future of DLLMs.

However, their adoption is currently challenged by performance issues in reasoning tasks where tokens are statistically dependent (Li et al., 2024; Xu et al., 2025; Feng et al., 2025; Wu et al., 2025; Liu et al., 2025; Song & Zhou, 2025; Kim et al., 2025). To see this, consider the example discussed by Song & Zhou (2025) and re-iterated by Wu et al. (2025): we want to predict the next two words/tokens given the context "*A poker hand that consists of two English words is:* _ _". Suitable predictions are "high card," "two pair," "full house," or "straight flush." Importantly, a strong dependence exists between these two words. However, concurrent prediction in DLLMs does not consider this dependence. This is because prediction in recent DLLMs (Nie et al., 2025; Yang et al., 2025; You et al., 2025) based on masked diffusion modeling (MDM) (Sahoo et al., 2024; Shi et al., 2024; Zheng et al., 2025; Ou et al., 2025) uses a deep net to compute probability distributions over the vocabulary for each of the masked tokens, and samples from these distributions independently when predicting concurrently. In the poker example, each of the first words "high," "two," "full," and "straight" will have roughly a 1/4 probability of occurring. Similarly, the second words "card," "pair," "house," and "flush" also have roughly a 1/4 probability of occurring. Due to independent sampling, we are more likely to predict undesirable results than a desirable outcome.

To address this challenge we propose to model a joint token distribution during concurrent prediction by introducing latent variables. A latent variable model enables us to capture any arbitrary joint distribution across tokens, as opposed to one that factorizes. Intuitively, following classic graphical model literature, conditioned on the latent variable we can sample tokens independently, yet, when marginalizing over the latent variable, *i.e.*, when sampling many times, we obtain samples from the proper joint distribution if the model is trained well. This approach follows the classic variational

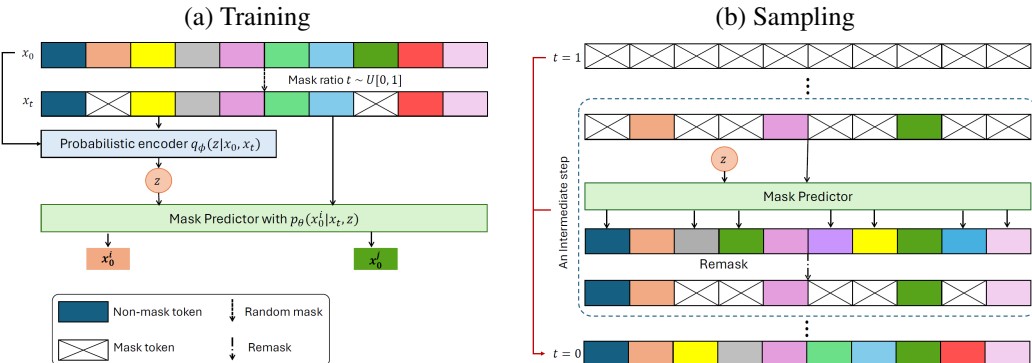

Figure 1: Conceptual overview of VMD. (a) Training. The encoder and mask predictor are trained on text with random masks applied independently to all tokens at the same ratio $t \sim U[0, 1]$. $x_0^i$ and $x_0^j$ are conditionally independent given the latent variable $z$ (b) Sampling. Our VMD uses the latent sample $z$ to achieve concurrent mask prediction and recover all tokens at $t = 0$ from the fully masked sequence at $t = 1$ with a flexible remasking strategy.

inference paradigm underlying expectation maximization or variational autoencoders. Hence, we refer to this framework as "Variational Masked Diffusion" (VMD).

We validate the effectiveness of VMD through controlled experiments on synthetic datasets, as well as on Sudoku puzzles and text datasets, where learning of dependencies among tokens improves global consistency. Across these domains, VMD enhances both generation quality and dependency awareness, highlighting the value of integrating variational inference into masked diffusion.

## 2 PRELIMINARIES

Diffusion models for discrete data were explored recently (Hoogeboom et al., 2021; Austin et al., 2023; Campbell et al., 2022a), and various noise processes were studied. Masked Diffusion Models (MDM) (Sahoo et al., 2024; Shi et al., 2024; Zheng et al., 2025; Ou et al., 2025), also referred to as absorbing state discrete diffusion models, have gained considerable attention. MDMs use a forward noising process where the original data sequence $x_0 = (x_0^1, \ldots, x_0^n)$ consists of $n$ data tokens at diffusion time $t = 0$ which are progressively replaced by a special [MASK] token as diffusion time $t$ increases to $t = 1$. Using 'Cat' to refer to a categorical distribution, this process is defined by the following transition probability:

$$q_{t|0}(x_t|x_0) = \prod_{i=1}^{n} q_{t|0}(x_t^i|x_0^i) = \prod_{i=1}^{n} \text{Cat}(x_t^i; (1-t)\delta_{x_t^i} + t\delta_{\texttt{[MASK]}}). \quad (1)$$

Training of an MDM, as has been shown recently (Sahoo et al., 2024; Shi et al., 2024; Zheng et al., 2025; Ou et al., 2025), makes use of an objective that can be directly derived from the data likelihood. This leads to the following principled evidence lower bound (ELBO) on the data log likelihood $\log p_\theta(x_0)$:

$$-\log p_\theta(x_0) \leq \int_0^1 \frac{1}{t} \mathbb{E}_{q_{t|0}(x_t|x_0)} \left[ \sum_{i:x_t^i = \texttt{[MASK]}} -\log p_\theta\left(x_0^i|x_t\right) \right] dt. \quad (2)$$

This loss or similar variants are typically used for training a model $p_\theta(x_0^i|x_t)$, which predicts a categorical distribution over the vocabulary for token $x_0^i$ conditioned on partially masked input $x_t$.

Inference, *i.e.*, the reverse process of the masking defined in Equation (1), is computationally inefficient as it involves modifying only one token per step (Campbell et al., 2022a; Lou et al., 2024). It is common to apply $\tau$-leaping (Gillespie, 2001). This enables us to concurrently unveil multiple masked tokens in a single step from noise level $t$ to noise level $s < t$.

However, as illustrated in Section 1 using the poker example, $\tau$-leaping (Gillespie, 2001) in diffusion-based large language models faces a trade-off: consider dependence among tokens by decoding one token at a time which is slow, or ignore dependence. This trade-off is suboptimal.

**Algorithm 1:** VMD Training

**Input** : Dataset $\mathcal{D}$, encoder $\mu_\phi$ and $\sigma_\phi$, decoder $p_\theta$.

1 **while** *stopping conditions not satisfied* **do**
2     Sample $t \sim \mathcal{U}[0, 1]$;
3     Sample $x_0 \sim \mathcal{D}$;
4     Sample $x_t \sim q_{t|0}(x_t|x_0)$;
5     $z = \mu_\phi(x_t, x_0) + \epsilon\sigma_\phi(x_t, x_0)$ with $\epsilon \sim \mathcal{N}(0, I)$;
6     Compute $\mathcal{L}_{\text{VMD}}$ in Equation (6);
7     Perform gradient update on $\theta$ and $\phi$;
8 **end**
**Output:** $\theta$ and $\phi$

**Algorithm 2:** VMD Sampling

**Input** : Decoder $p_\theta$, fully or partially masked input sequence $x$.

1 $z \sim \mathcal{N}(0, I)$, $x_1 = x$;
2 **for** $n = N$ **to** 1 **do**
3     $t_n = \frac{n}{N}, t_{n-1} = \frac{n-1}{N}$, and $x_{t_{n-1}} = x_{t_n}$;
4     $\hat{x}_0^i = \text{argmax}_v \, p_\theta(x_0^i = v|x_{t_n}, z), \forall i : x_{t_n}^i = [\text{MASK}]$;
5     $x_{t_{n-1}}^i = \hat{x}_0^i, \forall i : x_{t_n}^i = [\text{MASK}]$ and is not remasked;
6 **end**
**Output:** $x_0$

## 3   VARIATIONAL MASKED DIFFUSION (VMD)

To improve this trade-off, we first discuss in Section 3.1 a basic formulation to introduce a latent variable. We then expand this basic variational formulation to block diffusion in Section 3.2. We illustrate the benefits of the basic formulation and block diffusion in controlled experiments on synthetic data in Sections 4.1 and 4.2. We study more complex data in Sections 4.3 and 4.4.

### 3.1   BASIC VARIATIONAL FORMULATION

Our basic formulation is contrasted to classic MDM in Figure 1. Specifically, we introduce a global latent variable. This latent variable permits us to properly characterize arbitrary multi-modal joint probability distributions. We first define the joint conditional distribution over the entire sequence:

$$p_\theta(x_s|x_t) = \int \prod_{i=1}^{n} p_\theta(x_s^i|x_t, z)p(z)dz. \tag{3}$$

The marginal distribution for a single coordinate $x_0^i$ then becomes:

$$p_\theta(x_0^i|x_t) = \int p_\theta(x_0^i|x_t, z)p(z)dz. \tag{4}$$

Note that the latent variable $z$ is global, *i.e.*, it does not depend on the token position $i$. This is important because it enables us to model a joint distribution across tokens. Said differently and following classic graphical model literature: conditioned on the latent variable we can sample tokens $i$ and $j$ independently, yet, when marginalizing over the latent variable we obtain samples from the proper joint distribution if the model is trained well. This is formalized as follows:

$$p_\theta(x_s^i, x_s^j|x_t) = \int p_\theta(x_s^i|x_t, z) \cdot p_\theta(x_s^j|x_t, z)p(z)dz. \tag{5}$$

Incorporating the model proposed in Equation (4) into the training objective of MDM, given in Equation (2) yields

$$-\log p_\theta(x_0) \leq \mathcal{L}_{\text{VMD}}$$
$$\triangleq \int_0^1 \frac{1}{t} \mathbb{E}_{q_{t|0}(x_t|x_0)} \left[ \mathbb{E}_{q_\phi} \left[ \sum_{i:x_t^i=[\text{MASK}]} -\log p_\theta(x_0^i|x_t, z) \right] + D_{\text{KL}}\left(q_\phi(\cdot|x_0, x_t)||p(\cdot)\right) \right] dt. \tag{6}$$

Here, $p(\cdot)$ is a prior distribution over the latent space, *e.g.*, a standard Gaussian. Moreover, $q_\phi(\cdot|x_0, x_t)$ is an approximate posterior parameterized by trainable parameters $\phi$. In a variational autoencoder setting it is often referred to as the encoder while $p_\theta$ is called the decoder. As the approximate posterior is dataset dependent we provide architecture details in Section 4 and Appendix D. The derivation of the negative evidence lower bound (NELBO) is deferred to Appendix A. Note that Equation (6) maintains the evidence lower bound property.

**Algorithm 3:** Block VMD Training

**Input** : Dataset $\mathcal{D}$, encoder $\mu_\phi$ and $\sigma_\phi$, decoder $p_\theta$, number of blocks $B$.

1 **while** *stopping conditions not satisfied* **do**

2 $\quad$ Sample $\{t^b\}_{b=1}^B \sim \mathcal{U}[0,1]$;

3 $\quad$ Sample $\{x_0^b\}_{b=1}^B \sim \mathcal{D}$;

4 $\quad$ Sample $\{x_t^b\}_{b=1}^B \sim q_{t^b}(x_t^b|x_0^b)$;

5 $\quad$ $z^b = \mu_\phi\left(x_t^b, x_0^{\leq b}\right) + \sigma_\phi\left(x_t^b, x_0^{\leq b}\right)\epsilon$ with $\epsilon \sim \mathcal{N}(0,I)$, $\forall b$ ;

6 $\quad$ Compute $\mathcal{L}_{\text{BVMD}}$ in Equation (8);

7 $\quad$ Perform gradient update on $\theta$ and $\phi$;

8 **end**

**Output:** $\theta$ and $\phi$

**Algorithm 4:** Block VMD Sampling

**Input** : Decoder $p_\theta$, number of blocks $B$, fully or partially masked input sequence $x$.

1 **for** $b = 1$ **to** $B$ **do**

2 $\quad$ $z^b \sim \mathcal{N}(0,I)$, $x_1^b = x^b$;

3 $\quad$ **for** $n = N$ **to** $1$ **do**

4 $\quad\quad$ $t_n = \frac{n}{N}$, $t_{n-1} = \frac{n-1}{N}$, and $x_{t_{n-1}}^b = x_{t_n}^b$;

5 $\quad\quad$ $\hat{x}_0^{b,i} = \text{argmax}_v\, p_\theta(x_0^i = v|x_{t_n}^b, x_0^{<b}, z^{\leq b})$ $\quad \forall i: x_t^{b,i} = [\text{MASK}]$;

6 $\quad\quad$ $x_{t_{n-1}}^{b,i} = \hat{x}_0^{b,i}, \forall i: x_{t_n}^{b,i} = [\text{MASK}]$ and is not remasked;

7 $\quad$ **end**

8 $\quad$ $x_0 \leftarrow x_0^{1:b-1} \oplus x_0^b$;

9 **end**

**Output:** $x_0$

For inference, we draw the latent variable $z$ from a standard Gaussian. The trained decoder $p_\theta$ is used to predict all masked tokens at diffusion time $t$ with $\hat{x}_0^i = \text{argmax}_v\, p_\theta(x_0^i = v|x_t, z)$ for all $i : x_t^i = [\text{MASK}]$. Here, $v$ is a value in the vocabulary. Based on the remasking rules detailed in Section 3.3, we remask a portion of the tokens and iterate until all tokens are predicted.

We summarize the procedure for training and inference in Algorithm 1 and Algorithm 2, respectively. Training differs from classic masked diffusion model training in minimizing both a cross entropy term and a KL divergence between the approximate posterior and the prior. Inference differs from classic masked diffusion model inference in sampling of a single latent variable $z$ for the whole input sequence.

## 3.2 BLOCK DIFFUSION FORMULATION

While a latent variable enables to model dependencies among tokens more accurately, it is challenging to scale this across a large number of tokens. To address this, we follow recent research and combine the strengths of both diffusion and autoregressive models.

A prime example is the Block Diffusion Language Model (BD3-LM) (Arriola et al., 2025). This model defines an autoregressive probability distribution over blocks of discrete random variables, with the conditional probability of a block given previous blocks specified by a discrete denoising diffusion model. This hybrid approach effectively overcomes the limitations of both pure diffusion and autoregressive models by supporting flexible-length generation, a historical challenge for DLLMs, and significantly improves inference efficiency by incorporating KV caching and parallel token sampling. However, within each block, BD3-LM pursues classic unmasking, *i.e.*, it does not consider the dependencies among tokens that are predicted concurrently.

To address this, we use a hybrid approach that models blocks of tokens autoregressively and applies the variational diffusion formulation discussed in Section 3.1 within each block.

Formally, the tokens $x$ are grouped in $B$ consecutive blocks, each of length $r$, with total sequence length $L = Br$. For readability, we use $x^b$ to refer to the block containing tokens at positions $(b-1)r + 1$ to $br$ with $b \in \{1, \ldots, B\}$. We use a latent variable $z^b$ for each block. Using this notation, the data log-likelihood can be factorized as follows:

$$\log p_\theta(x_0|x_t) = \sum_{b=1}^B \log p_\theta(x_0^b|x_t^b, x_0^{<b}) = \sum_{b=1}^B \log \int p_\theta(x_0^b|x_t^b, x_0^{<b}, z^{\leq b}) p(z^{\leq b}) dz^{\leq b}, \quad (7)$$

where $p(z^{\leq b})$ denotes the prior distribution of the latent $z^{\leq b}$. The Negative Evidence Lower Bound (NELBO) for the data log-likelihood in Equation (6) is changed to

$$
-\log p_\theta(x_0) \leq \mathcal{L}_{\text{BVMD}} \triangleq \sum_{b=1}^{B} \int_0^1 \frac{1}{t} \mathbb{E}_{q_{t|0}(x_t|x_0)} \left[ \mathbb{E}_{q_\phi} \left[ \sum_{i:x_t^{b,i}=[\text{MASK}]} -\log p_\theta\left(x_0^{b,i}|x_t^b, x_0^{<b}, z^{\leq b}\right) \right] \right.
$$

$$
\left. + D_{\text{KL}}\left(q_\phi\left(\cdot|x_t^b, x_0^{\leq b}\right) \| p\left(\cdot\right)\right)\right] dt, \tag{8}
$$

where $q_\phi\left(\cdot|x_t^b, x_0^{\leq b}\right)$ is the approximate posterior for the latent parameterized by a neural network with trainable parameters $\phi$. We use a simple prior $p\left(z^{\leq b}\right) = \mathcal{N}(z^{\leq b}; 0, I)$. The derivation of the bound in Equation (8) is deferred to Appendix B.

Note, when the number of blocks $B = 1$ ($r = L$), the model discussed here (Equation (7)) is identical to the model detailed Equation (4). Moreover, when $B = L$ ($r = 1$), we obtain an autoregressive variational model. Therefore, varying $B$ (or equivalently $r$) allows us to interpolate between an autoregressive variational model and the variational masked diffusion model.

We summarize training and inference for the variational block diffusion formulation in Algorithm 3 and Algorithm 4 respectively. During training, we adopt a fully vectorized approach where noisy inputs $x_t$ and clean data $x_0$ are concatenated and processed jointly by the model. An attention mask enforces the dependency structure so that the encoder, when computing the latent $z^b$, only depends on $(x_t^b, x_0^{\leq b})$. In contrast, the decoder $p_\theta$ depends on $(x_t^b, x_0^{<b}, z^{\leq b})$.

During inference, we follow block diffusion (Arriola et al., 2025) and apply KV caching for efficient sampling. Each generated block $x_0^b$ is stored in the cache, which means that the prefix context $x_0^{<b}$ used during training corresponds to the accumulated keys and values $(K^{1:b-1}, V^{1:b-1})$. The decoder prediction for block $b$, therefore, depends only on the current noisy input $x_t^b$, the latent $z^{\leq b}$, and the cached context from earlier generated blocks. In short, training uses ground-truth prefixes while inference reuses cached context, which ensures both correctness and efficiency.

### 3.3 REMASKING

Early DLLMs (Nie et al., 2025) remask the predictions with the lowest confidence, *i.e.*, those with the lowest $p_\theta$ values. The remasking strategies considered here include: random remasking and confidence-based remasking/unmasking (Nie et al., 2025; Zheng et al., 2023; Kim et al., 2025). The confidence scores used in prior work do not consider the dependence between tokens: they are token-local metrics. Differently, our variational models provide confidence scores with more global context through the latent variables. Specifically, we consider the following two strategies:

The first strategy is to check the probability of the selected value at each position (Nie et al., 2025; Zheng et al., 2023). Concretely, if $v_1$ is the most probable value in the vocabulary according to $p_\theta(x_0^i|x_t, z)$, then the confidence for token $i$ is defined as $c_{\text{prob.}}^i = p_\theta(x_0^i = v_1|x_t, z)$.

The second strategy uses the top-$K$ probability margin (Kim et al., 2025). The uncertainty of a token is estimated using the absolute difference between the two most probable values at position $i$. If $v_1$ and $v_2$ are the two most probable values in vocabulary according to $p_\theta(x_0^i|x_t, z)$, the confidence of token $i$ is given by $c_{\text{marg.}}^i = |p_\theta(x_0^i = v_1|x_t, z) - p_\theta(x_0^i = v_2|x_t, z)|$.

To extend both strategies to block VMD, we replace $p_\theta(x_0^i|x_t, z)$ with $p_\theta(x_0^{b,i}|x_t^b, x_0^{<b}, z^{\leq b})$.

## 4 EXPERIMENTS

We first study efficacy of the proposed Variational Masked Diffusion (VMD) on controlled synthetic data in Sections 4.1 and 4.2. This enables us to carefully assess and visualize the asserted properties of the different formulations. We then present results on Sudoku data in Section 4.3, which has stronger dependencies between tokens than classic text data. Finally, we also present results on text data in Section 4.4, demonstrating that the proposed VMD formulation is on par with prior work.

Table 1: Results for controlled synthetic data with 2 tokens. We report KL divergence (↓) and accuracy (↑). While baseline models fail in one-step inference and degenerate to random guessing, VMD successfully captures token dependencies, yielding substantially higher accuracy and lower KL divergence. Best result highlighted in **bold** font.

| Experiment | Inference | MDM (KL ↓) | MDM (Acc ↑) | VMD (KL ↓) | VMD (Acc ↑) |
|---|---|---|---|---|---|
| Deterministic | One-step | 2.3 | 10.18% | **0.082** | **93.2%** |
| | Token-by-token | **0.001** | 99.97% | 0.012 | **100%** |
| Non-Uniform | One-step | 2.2 | 11.63% | **0.081** | **93.04%** |
| | Token-by-token | 0.023 | 99.96% | **0.009** | **99.98%** |

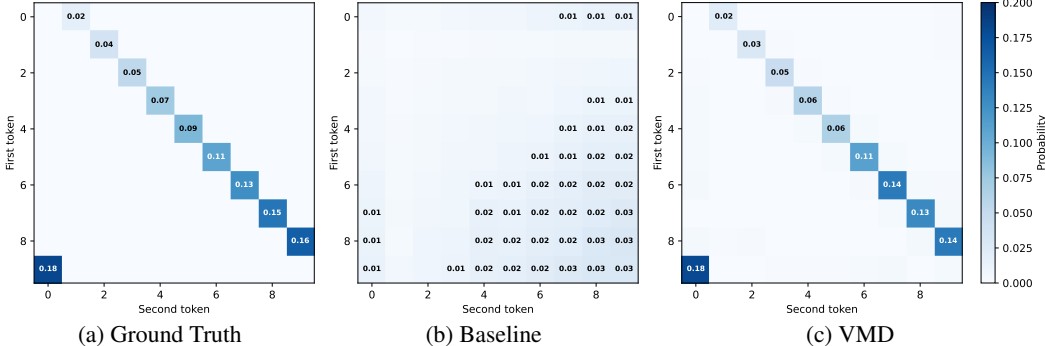

(a) Ground Truth     (b) Baseline     (c) VMD

Figure 3: Results on controlled synthetic data with 2 tokens under the non-uniform setting. (a) Ground truth distribution, (b) baseline masked diffusion with one-step generation, and (c) our VMD with one-step generation. While the baseline degenerates into nearly uniform random guessing and fails to capture the underlying dependency, VMD accurately recovers the true distribution, demonstrating its ability to learn token-to-token correlations beyond conditional independence.

## 4.1 CONTROLLED SYNTHETIC DATA WITH 2 TOKENS

To examine whether models can capture dependency among tokens, we first evaluate on controlled synthetic datasets where the ground truth distribution is fully specified and easy to visualize. All sequences contain only two tokens, which makes it straightforward to measure whether a model has learned the correct dependency structure. The baseline is a block diffusion model with a small number of DiT blocks. For VMD, the decoder backbone is similar to the baseline to ensure equal inference cost, with an additional lightweight pathway to incorporate latent variables. The encoder uses the same architecture with one quarter of the DiT blocks. A 32-dimensional latent variable is injected into the decoder. See Appendix C.1 for more implementation details. We report two metrics at different numbers of function evaluations (NFE): (i) accuracy, the proportion of valid sequences among generated samples, and (ii) KL divergence, the distance between the generated distribution and the ground truth distribution.

**Deterministic dependency.** In the first setting, we consider the following 10 two-token sequences: $\{(k, k+1 \mod 10)\}_{k=0}^{9}$ and the data distribution is uniform on the support. Here, the second token is fully determined by the first. From Table 1, we see that standard masked diffusion fails to capture this dependency when decoding concurrently (one-step inference): when generating both tokens simultaneously, it degenerates to random guessing yielding around 10% accuracy. In contrast, VMD is able to capture and generate correct pairs reliably. This shows that a latent variable enables the model to represent dependencies among tokens.

**Non-uniform distribution.** Next, we consider a non-uniform distribution over the support, with $P((k, k+1 \mod 10)) = \frac{k+1}{55}$ for $k \in \{0, ..., 9\}$. While both models

Figure 2: KL divergence of experiments on data with varying dependence strength for one-step generation. Higher $p$ values indicate stronger dependence among tokens.

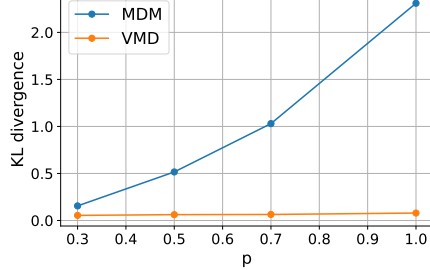

Table 2: Accuracy (↑) (first two rows) and KL divergence (↓) (last two rows) for the generated sequences when starting from fully masked tokens. Sequence length $L = 4$. Column '$B = 2$, NFE $= 4$': unmasking one token at a time within each block and autoregressive decoding from block 1 to block 2; Column '$B=2$, NFE $= 2$': parallel decoding within a block and autoregressive among blocks; Column '$B=1$, NFE $= 4$': recovering the sequence token-by-token; Column '$B=1$, NFE $= 1$': simultaneous recovery of all tokens. **Bold** for the best.

| | | D1 | | | | D2 | | | |
|---|---|---|---|---|---|---|---|---|---|
| | Method | $B=2$ NFE$=4$ | $B=2$ NFE$=2$ | $B=1$ NFE$=4$ | $B=1$ NFE$=1$ | $B=2$ NFE$=4$ | $B=2$ NFE$=2$ | $B=1$ NFE$=4$ | $B=1$ NFE$=1$ |
| Acc. (↑) | Block MDM | 99.4% | 10.1% | 99.3% | 0.1 % | 99.3% | 1.1% | 99.1 | 1.1% |
| | VMD (ours) | **100%** | **97.8%** | **100%** | **93.3%** | **100%** | **96.3 %** | **100.0%** | **94.1 %** |
| KL (↓) | Block MDM | **0.007** | 2.298 | **0.010** | 9.422 | 0.018 | 7.656 | **0.015** | 7.493 |
| | VMD (ours) | 0.012 | **0.045** | 0.029 | **0.099** | **0.017** | **0.053** | 0.475 | **0.093** |

achieve near-perfect accuracy with token-by-token generation, KL divergence and heatmap results in Table 1 and Figure 3 reveal that VMD samples more faithfully from the target distribution. This indicates that VMD not only captures dependencies among tokens but also accurately models the underlying data distribution.

**Varying dependence strength.** Finally, we interpolate between dependent and random pairs. Given a token $x_1$, the second token is set to $(x_1+1) \mod V$ with probability $p$, and to a random alternative with equal probability $\frac{1-p}{V-1}$ otherwise, where $V$ is the vocabulary size. As $p$ decreases from 1 to 0, the token dependency transitions from fully dependent to independent. In this experiment, we cannot use accuracy since every pair of tokens is valid. Therefore, we only report the KL divergence and show heatmaps (Appendix D.1). Results in Section 4.1 show that VMD maintains accurate distributional modeling even if the dependence is weak.

Taken together, the three controlled experiments demonstrate: (i) for strongly dependent tokens, VMD yields substantial one-step generation improvements; (ii) in probabilistic settings, VMD more faithfully models the underlying distribution; and (iii) across the full spectrum from strong to weak dependence, VMD consistently models the true data distribution accurately. These results show that VMD successfully learns the token dependency, validating its design. Further experimental results can be found in Appendix D.1.

## 4.2 CONTROLLED SYNTHETIC DATA WITH 4 TOKENS

To further evaluate whether VMD can scale from simple pairwise dependencies to longer sequences, we construct synthetic data with four tokens per sequence. This setting allows us to introduce block structure explicitly and test whether latent variables can simultaneously capture intra-block correlations and preserve cross-block dependencies. We adopt the same model architecture and evaluation metrics as in the two-token experiments, and set the block size to 2, so that each sequence of length 4 is divided into two blocks. Each block is assigned a 32-dimensional latent variable.

The first dataset (D1) contains 10 unique sequences: $\{(k, k+1, k+2, k+3 \mod 10)\}_{k=0}^9$, which have strong token dependence. The second dataset (D2) is $\{(k, k+1, l, l+1 \mod 10)\}_{k,l=0}^9$. For D2, the first block of length 2 is independent of the second block of length 2, and the total number of unique sequences is 100. Note that if we independently draw a number for each location, then the total number of unique sequences is $10^4$. Therefore, if we concurrently unmask the digits independently (in one step), the success rate for D1 is $0.1\%$ and the success rate for D2 is $1.0\%$ in theory. We provide the results in Section 4.2 and observe: when decoding one token at a time, both classic block diffusion as well as our variational formulation achieve almost 100% prediction accuracy. However, when decoding all four tokens in parallel ($B = 1$, NFE $= 1$), block diffusion behaves like classic MDM and can't do better than random guessing (0.1% for D1, and 1.1% for D2), while our approach achieves a much better accuracy of 81.5% for D1 and 64.4% for D2. When setting $B=2$ and NFE $= 2$, we see that the accuracy of block diffusion increased from $0.1\%$ to $10\%$ for D1. The accuracy didn't change much for D2, since the two consecutive blocks are independent. Our method's accuracy further increased to 91.3% for D1 and 72.8% for D2.

Table 3: Sudoku results for different sampling methods and NFE values. VMD consistently improves over the baseline across both sampling schemes. Best results shown in **bold** font.

| Model | Top prob (Accuracy ↑) | | | Top prob margin (Accuracy ↑) | | |
|---|---|---|---|---|---|---|
| | NFE=5 | NFE=10 | NFE=20 | NFE=5 | NFE=10 | NFE=20 |
| Baseline | 10.6% | 14.7% | 20.4% | 36.2% | 78.4% | 91.1% |
| VMD | **67.7%** | **76.4%** | **80.9%** | **96.9%** | **99.0%** | **99.7%** |

These results show: VMD extends naturally to block-structured settings. With one latent per block, VMD captures token correlations within blocks, preserves cross-block dependencies when they exist, and avoids inventing dependencies when they do not. Hence, the variational formulation not only addresses the limitations of standard masked diffusion but also integrates seamlessly with block diffusion to provide a principled and flexible framework for modeling dependencies across scales.

## 4.3 SUDOKU DATA

Sudoku is a classical logic-based puzzle played on a $9 \times 9$ grid. The objective is to fill empty cells such that every row, column, and each of the nine $3 \times 3$ subgrids contains all digits from 1 to 9 exactly once. This formulation naturally imposes global and local dependencies among numbers. The validity of each digit depends not only on its immediate neighbors but also on distant constraints within the same row, column, or subgrid. This makes Sudoku a good benchmark for assessing whether generative models can capture token dependencies.

**Setup.** We follow the experimental pipeline established by Kim et al. (2025). Specifically, their work adopts the model and training procedure introduced by Ye et al. (2025), but trains on the dataset of Shah et al. (2024). The data consists of 1.8M training puzzles and 0.1M test puzzles. Compared with the smaller 1M-puzzle dataset used by Ye et al. (2025), the benchmark of Shah et al. (2024) is substantially more challenging and provides a more rigorous test of model generalization. We adopt the same setting and their model as a baseline to ensure comparability and robustness of the evaluation. Following prior work, we represent Sudoku puzzles by flattening each $9 \times 9$ grid into a sequence of 81 digits. Entries to be filled are denoted by the digit 0. During training, we concatenate the puzzle and its solution, separated by special tokens, yielding an input sequence of length 164. Unlike our other experiments, here we do not employ any block structure.

**Implementation Details for VMD.** The baseline model contains 5.5M parameters. To guarantee comparable inference speed, for our decoder backbone we use a similar size as the baseline, while adding a lightweight module to incorporate latent variables. The resulting model has a total of 5.2M parameters. For the encoder, we use the same architecture as the decoder but reduce the number of Transformer layers from 6 to 4. The input sequence is associated with a 128-dimensional latent variable, which is embedded and injected into every DiT block in the decoder via a shared adaptive layernorm module following the design of Chen et al. (2025), providing a compact yet expressive representation of dependencies across tokens.

**Evaluation and Results.** For clarity, we briefly recall the two confidence-based remasking strategies used in Table 3. Given the per token distribution $p_\theta(x_0^i|x_t, z)$, the top probability rule selects the position whose most likely token $v_1$ has the largest absolute probability, *i.e.*, $c_{\text{prob.}}^i = p_\theta(x_0^i = v_1|x_t, z)$. The top probability margin rule instead selects the position where the model shows the strongest relative preference between the top two tokens. This is measured by $c_{\text{marg.}}^i = |p_\theta(x_0^i = v_1|x_t, z) - p_\theta(x_0^i = v_2|x_t, z)|$. We evaluate both strategies because they capture different types of model confidence and lead to different decoding behaviors. We evaluate the performance using the percentage of puzzles that are completely solved (*i.e.*, every cell is correct). This metric directly reflects the ability of a generative model to learn the dependence among tokens, inherent in Sudoku. As shown in Table 3, we successfully reproduce the baseline accuracy reported by Kim et al. (2025). Our VMD model consistently outperforms the baseline using confidence-based remasking with $c_{\text{prop.}}$ and $c_{\text{marg.}}$. The gains are particularly pronounced at lower NFE, highlighting that VMD can generate valid solutions more efficiently. Since both approaches use the same training data and backbone architecture, these results provide strong evidence that VMD learns dependence among tokens that standard masked diffusion fails to capture, directly supporting our claim.

## 4.4 TEXT DATA

Text data is the most widely used and extensively studied data modality in generative modeling. We evaluate VMD on two representative datasets. The first is the text8 dataset (Hutter, 2006), which consists of the first 100M characters from Wikipedia. The dataset is preprocessed into a character-level corpus containing 26 English letters and the space symbol, yielding a vocabulary size of 27. Its simplicity and popularity make it a standard benchmark for discrete sequence modeling. The second is the One Billion Word (LM1B) dataset (Chelba et al., 2014), a widely used real-world text corpus of about 30M sentences. Together, these two datasets allow us to study VMD across both controlled character-level settings and more realistic language modeling scenarios.

Table 4: Test perplexity (PPL; ↓) on text8 dataset. **Bold** for the best.

|  | **PPL** ($\downarrow$) |
| --- | --- |
| Autoregressive | 2.603 |
| SEDD | $\leq 3.529$ |
| MDLM | $\leq 3.498$ |
| BD3-LM Block size 4 | $\leq 2.873$ |
| BD3-LM Block size 8 | $\leq 3.126$ |
| VMD Block size 4 | $\leq$ **2.858** |
| VMD Block size 8 | $\leq 3.125$ |
| VMD Block size 256 | $\leq 3.405$ |

**Implementation Details for VMD.** For text8, we segment sequences into fixed-length chunks of 256 tokens and experiment with block sizes of 4 and 8 for block diffusion (Arriola et al., 2025) and our block VMD. For LM1B, we segment sequences into 128 token chunks and evaluate with a block size of 4. For both encoder and decoder, we adapt the DiT architecture of BD3-LM. To ensure similar inference speed, the decoder backbone is kept identical to the baseline except for an additional lightweight module to process the latent variable. For the encoder, we reduce the number of layers so that its architecture mirrors the decoder while maintaining half of its parameter count. A 128-dimensional latent vector is assigned to each block, providing a compact representation of intra-block dependency, while blocks are processed autoregressively using Transformer layers.

**Results.** We evaluate VMD on both character-level and word-level language modeling tasks. On the text8 dataset, we compare our method against two state-of-the-art families: (i) strong autoregressive models trained under comparable conditions, and (ii) recent diffusion-based language models, including SEDD (Lou et al., 2024), MDLM (Sahoo et al., 2024), and BD3-LM (Arriola et al., 2025). Test perplexities are reported in Table 4. VMD improves over recent diffusion-based models (SEDD, MDLM) and provides consistent gains over BD3-LM at block sizes 4 and 8. Notably, we also evaluate a block size of 256, which removes the block structure entirely and corresponds to a single block. Even in this setting, VMD outperforms all diffusion baselines without block structure, indicating that the latent variables introduced by VMD are intrinsically effective and not dependent on block structure. Among diffusion-based methods, VMD with a block size of 4 achieves the best perplexity on text8.

We further evaluate VMD on LM1B. VMD again improves over BD3-LM in test perplexity, generative perplexity, and yields higher sample entropy. Generative perplexity and entropy are evaluated by a pretrained GPT-2 Large model using 256 samples. These results show that VMD provides a valuable extension of masked diffusion and leads to stronger performance on both character level and word level natural language data.

Table 5: Language modeling results on LM1B. VMD achieves lower test and generative perplexity and higher entropy than BD3-LM, showing that VMD models token dependencies more effectively.

| Model | Test PPL ↓ | Generative PPL ↓ | Entropy ↑ |
| --- | --- | --- | --- |
| BD3-LM (Block size 4) | 46.26 | $123.243 \pm 31.669$ | $4.149 \pm 0.415$ |
| VMD (Block size 4) | **44.88** | **$107.430 \pm 12.600$** | **$4.281 \pm 0.076$** |

## 5 RELATED WORK

**Discrete diffusion models.** (Continuous) diffusion models are based on continuous-space Markov chains with Gaussian transition kernels (Ho et al., 2020). In a similar spirit, discrete diffusion models have emerged from discrete-space Markov chains (Hoogeboom et al., 2021). Austin et al. (2021) introduced D3PM, which generalizes the multinomial diffusion model (Hoogeboom et al., 2021) by

going beyond corruption processes with uniform transition probabilities. D3PM formulates a large design space for discrete diffusion models, generalizing and enabling the exploration of new types of corruption processes, including those with absorbing states. The D3PM framework also drew insightful connections between diffusion with absorbing states and masked language models, serving as a critical bridge for the masked diffusion paradigm that would follow. Campbell et al. (2022b) provided the complete continuous time framework for denoising diffusion models of discrete data and incorporated the $\tau$-leaping approach for efficient reverse time sampling. Later, Lou et al. (2024) introduced SEDD, which incorporates a theoretically and practically robust score-entropy objective. Alternatively, several works proposed to perform diffusion over continuous embeddings of discrete tokens (Li et al., 2022; Dieleman et al., 2022; Chen et al., 2023). This allows use of continuous diffusion algorithms (Ho et al., 2020; Song et al., 2021). However, it is challenging to find a good encoding and decoding between discrete tokens and the continuous latent space (Li et al., 2022).

**Masked diffusion models.** Masked diffusion models are a powerful and popular sub-field of discrete diffusion models. Building upon the absorbing transition kernel, Shi et al. (2024); Sahoo et al. (2024) introduce *Masked Diffusion Models* (MDM), which have a simple and principled training recipe, with time dependent mask ratio. LLaDA (Nie et al., 2025) and LLaDA-V (You et al., 2025) scaled up the original MDMs and demonstrated their strengths in language modeling. However, all the existing masked diffusion models cannot capture the dependence among concurrently sampled tokens well. Our model addresses this by introducing a latent variable.

Arriola et al. (2025) introduce block discrete denoising diffusion language models (BD3-LMs), which interpolate between discrete diffusion and autoregressive models. It combines the strengths of autoregressive and diffusion models and enables variable-length, high-quality generation while enhancing inference efficiency through KV caching and parallel sampling. Our method further improves efficiency and accuracy of parallel sampling while using the block autoregressive structure.

**Other non-autoregressive models.** Efforts to model natural language in a non-autoregressive manner began with BERT (Devlin et al., 2019). Such non-causal methods utilize rich text representations. Extending these intuitions further, Hoogeboom et al. (2022) and Shih et al. (2022) introduced any-order modeling, enabling generation in arbitrary orders.

**Variational latent model.** Concurrently, Xie et al. (2025) introduce the variational autoencoding discrete diffusion model (VADD). It uses a latent continuous random variable to capture correlations among dimensions. The formulation is identical to our basic VMD in Section 3.1. However, in addition to the basic formulation, we introduce the variational block diffusion in Section 3.2. We also design synthetic examples to clearly illustrate the importance of using the latent variables and block structure to capture the dependencies among tokens. The latent variables are able to implicitly capture the dependencies, not just the correlations, among different tokens.

Kong et al. (2025) incorporate latent variables into autoregressive generation of tokens. Our approach differs as we study latent variables for masked diffusion.

## 6  CONCLUSION

We address a key limitation of classic masked diffusion modeling: how to consider the dependence among tokens when decoding concurrently. For this, we propose variational masked diffusion (VMD). It introduces a latent variable into masked diffusion modeling. On data where dependencies among tokens are crucial, VMD significantly improves results upon classic masked diffusion. On data where tokens are largely independent, the proposed formulation yields on par results. All code will be released to ensure reproducibility of the reported results.

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

APPENDIX: VARIATIONAL MASKED DIFFUSION MODELS

This appendix is structured as follows: in Section A we derive the variational masked diffusion (VMD) loss stated in Equation (6); in Section B we derive the block diffusion loss stated in Equation (8); in Section C we discuss implementation details; in Section D we provide additional experimental results; and in Section E we provide additional ablation studies.

## A  VMD LOSS DERIVATION

Here we provide the derivation of the VMD loss stated in Equation (6). For this we use the negative evidence lower bound (NELBO) for masked diffusion (Sahoo et al., 2024, Eq. (10)), *i.e.*,

$$-\log p_\theta(x_0) \leq \mathcal{L}_{\text{NELBO}}^\infty \triangleq \int_0^1 \mathbb{E}_{q_{t|0}(x_t|x_0)} \frac{\alpha_t'}{1-\alpha_t} \left[\log p_\theta(x_0|x_t)\right] dt$$

$$= \int_0^1 \frac{1}{t} \mathbb{E}_{q_{t|0}(x_t|x_0)} \left[-\log p_\theta(x_0|x_t)\right] dt, \quad \text{with} \quad \alpha_t = 1-t. \quad (9)$$

Next we derive the NELBO for $-\log p_\theta(x_0|x_t)$ as follows:

$$-\log p_\theta(x_0|x_t) = -\log \int p_\theta(x_0|x_t, z) p(z) dz$$

$$= -\log \int q_\phi(z|x_0, x_t) \frac{p_\theta(x_0|x_t, z) p(z)}{q_\phi(z|x_0, x_t)} dz$$

$$\leq \mathbb{E}_{q_\phi(z|x_0, x_t)} \left[-\log \frac{p_\theta(x_0^i|x_t, z) p(z)}{q_\phi(z|x_0, x_t)}\right]$$

$$= \mathbb{E}_{q_\phi(z|x_0, x_t)} \left[-\log p_\theta(x_0|x_t, z)\right] - \mathbb{E}_{q_\phi(z|x_0, x_t)} \left[\log \frac{p(z)}{q_\phi(z|x_0, x_t)}\right]$$

$$= \mathbb{E}_{q_\phi(z|x_0, x_t)} \left[-\log p_\theta(x_0|x_t, z)\right] + D_{\text{KL}} \left(q_\phi(\cdot|x_0, x_t)||p(\cdot)\right). \quad (10)$$

We assume $p_\theta(x_0|x_t, z) = \prod_{i=1}^L p_\theta(x_0^i|x_t, z)$. Plugging Equation (10) into Equation (9), we obtain

$$-\log p_\theta(x_0) \leq \mathcal{L}_{\text{NELBO}}^\infty$$

$$\leq \int_0^1 \frac{1}{t} \mathbb{E}_{q_{t|0}(x_t|x_0)} \left[\mathbb{E}_{q_\phi} \left[\sum_{i:x_t^i=\text{[MASK]}} -\log p_\theta(x_0^i|x_t, z)\right] + D_{\text{KL}} \left(q_\phi(\cdot|x_0, x_t)||p(\cdot)\right)\right] dt. \quad (11)$$

Here, $p(\cdot)$ is a prior distribution over the latent space, often chosen to be a standard Gaussian, *e.g.*, in a variational autoencoder setting. Moreover, $q_\phi(\cdot|x_0, x_t)$ is an approximate posterior parameterized by trainable parameters $\phi$.

# B    BLOCK VMD LOSS DERIVATION

According to Equation (7), we have

$$
\begin{aligned}
-\log p_\theta(x_0|x_t) &= -\sum_{b=1}^{B} \log \int p_\theta(x_0^b|x_t^b, x_0^{<b}, z^{\leq b}) p(z^{\leq b}) dz^{\leq b}, \\
&= -\sum_{b=1}^{B} \log \int q_\phi(z^{\leq b}|x_t^b, x_0^{\leq b}) \frac{p_\theta(x_0^b|x_t^b, x_0^{<b}, z^{\leq b}) p(z^{\leq b})}{q_\phi(z^{\leq b}|x_t^b, x_0^{\leq b})} dz^{\leq b} \\
&\leq -\sum_{b=1}^{B} \mathbb{E}_{q_\phi(z^{\leq b}|x_t^b, x_0^{\leq b})} \log \frac{p_\theta(x_0^b|x_t^b, x_0^{<b}, z^{\leq b}) p(z^{\leq b})}{q_\phi(z^{\leq b}|x_t^b, x_0^{\leq b})} dz^{\leq b} \\
&= \sum_{b=1}^{B} \mathbb{E}_{q_\phi(z^{\leq b}|x_t^b, x_0^{\leq b})} \left[ -\log p_\theta(x_0^b|x_t^b, x_0^{<b}, z^{\leq b}) - \log \frac{p(z^{\leq b})}{q_\phi(z^{\leq b}|x_t^b, x_0^{\leq b})} \right] \\
&= \sum_{b=1}^{B} \mathbb{E}_{q_\phi(z^{\leq b}|x_t^b, x_0^{\leq b})} \left[ -\log p_\theta(x_0^b|x_t^b, x_0^{<b}, z^{\leq b}) \right] + D_{\mathrm{KL}}(q_\phi(\cdot|x_t^b, x_0^{\leq b}) \| p(\cdot)).
\end{aligned}
$$

$$(12)$$

Here, $p(\cdot)$ is a prior distribution over the latent space, often chosen to be a standard Gaussian. Moreover, $q_\phi(\cdot|x_t^b, x_0^{\leq b})$ is an approximate posterior parameterized by trainable parameters $\phi$. Plugging Equation (12), into the masked diffusion NELBO given in Equation (9) yields the bound stated in Equation (8).

# C    IMPLEMENTATION DETAILS

## C.1    SYNTHETIC DATA

For the synthetic data experiments with both 2-token and 4-token sequences, we use a common model architecture adopted from BD3-LM (Arriola et al., 2025). The baseline is a block diffusion model with 8 DiT blocks and a hidden size of 64. For the proposed VMD, the decoder backbone remains identical to the baseline with an additional lightweight module to incorporate latent variables to ensure equal inference complexity, while the encoder uses only 2 DiT blocks with all other hyperparameters unchanged. Each block $x^b$ in the data sequence is associated with a 32-dimensional latent variable $z^b$. The latent is incorporated into the decoder as a conditioning signal following the adaLN-Zero formulation in DiT (Peebles & Xie, 2023). Within each DiT block, we first pass the latent embedding to an MLP and get 6 conditioning parameters $(\gamma_1, \beta_1, \alpha_1, \gamma_2, \beta_2, \alpha_2) = \mathrm{MLP}(z_{\mathrm{emb}})$. Then we inject these conditioning parameters within each DiT block as $h_1 = x_{\mathrm{emb}} + \alpha_1 \odot \mathrm{MSA}(\mathrm{LN}(x) \odot \gamma_1 + \beta_1)$ and $h_2 = h_1 + \alpha_2 \odot \mathrm{FFN}(\mathrm{LN}(h_1) \odot \gamma_2 + \beta_2)$. Here, MSA is a multi-head self-attention layer, LN is a layer norm, and FFN is a point-wise feed forward network.

Because the sequence lengths and block sizes are very small, we train with a large batch size of 10,000 for 2,000 steps. We use the Adam optimizer with a fixed learning rate of 1e-3. For the uniform data experiments, we set the KL weight to 4.0 to encourage the latent prior to match a uniform distribution. In all other experiments, when the KL weight is around 1.0, the performance is stable. Parameters are initialized with default PyTorch settings, and no additional regularization is applied beyond the KL term.

During inference, we adopt the top probability remasking strategy. The number of function evaluations (NFE) depends on the specific experiment and is reported in detail in Section 4.1 and Section 4.2. All experiments converge within a few minutes on a single NVIDIA L40S GPU.

## C.2    SUDOKU

For the Sudoku experiment, we follow Kim et al. (2025) to ensure a fair comparison by adopting the same dataset, model backbone, and training procedure. Specifically, the baseline uses the Hugging-

Face `AutoModelForCausalLM` with GPT-2 configuration ($n_{\text{layer}} = 3$, $n_{\text{head}} = 12$, embedding size 384). Our model backbone follows the DiT architecture employed by Arriola et al. (2025). For both the encoder and decoder, we adopt the tiny model configuration provided in their official GitHub repository. We set the encoder and decoder to have 4 and 6 DiT blocks respectively in order to match the model size of the baseline. We further introduce a 128-dimensional latent variable, which is embedded and then injected into each DiT block in the decoder, similar to the process for synthetic data. The 6 conditioning parameters ($\gamma_1, \beta_1, \alpha_1, \gamma_2, \beta_2, \alpha_2$) are shared across all DiT blocks. All other computations ($h_1, h_2$) remain identical to the ones used for synthetic data. This design (Chen et al., 2025) reduces the number of learnable parameters while preserving the expressive conditioning effect of the latent variable.

Data is represented by flattening each Sudoku puzzle into a sequence of 81 tokens. During training, the puzzle and its solution are concatenated into a 164-token sequence (81 puzzle + [SEP] token + 81 solution + [EOS] token), and the model is trained to predict the answer portion conditioned on the puzzle portion. This design implies that the latent variable cannot be obtained by taking a simple mean over the entire sequence dimension. Instead, we compute a learned weighted average over hidden states. First, a gating network produces a score for each position, and positions corresponding to the puzzle are masked out. The remaining scores are normalized with a softmax to produce attention weights, and finally, the latent is obtained as a weighted sum of hidden states according to these weights.

Training is performed with batch size 1,024, learning rate 1e-3, cosine learning rate scheduler, and 300 epochs. The dataset consists of 1.8M Sudoku puzzles for training and 0.1M for evaluation. During inference, only the puzzle tokens are provided, and the model generates the answer tokens with the remasking strategy described in the main text.

### C.3 Text

For the text experiments, we adopt a block structure and use the BD3-LM (Arriola et al., 2025) architecture as our baseline. The model processes sequences of length 256 with 8 DiT blocks and 8 attention heads. Our VMD decoder backbone is kept identical to this baseline to ensure equal inference cost, while the encoder mirrors the same architecture but uses only 4 DiT blocks, with all other parameters unchanged. Each block $b$ in the sequence is assigned a 128-dimensional latent variable $z^b$, which is embedded, scaled, and then added to the data embedding in the same manner as in the synthetic experiments.

Training is performed with batch size 512 using the AdamW optimizer and a learning rate of 3e-4. We employ a constant learning rate schedule with 2,500 warm-up steps.

## D ADDITIONAL EXPERIMENTAL RESULTS

### D.1 CONTROLLED SYNTHETIC DATA WITH 2 TOKENS

We present additional experimental results here. Figure 4 and Figure 5 present results on controlled synthetic data with 2 tokens under deterministic and varying token dependency settings. In the deterministic setting, the baseline degenerates into nearly uniform predictions and fails to recover the strong token dependence. In contrast, VMD closely matches the ground-truth distribution. In the varying token dependence setting, where a parameter $p$ controls the dependency strength, the baseline again produces noise-like outputs that are insensitive to the true dependency structure. In contrast, VMD adapts to both weak and strong dependence and successfully reproduces the ground-truth distributions. Together, these experiments demonstrate that VMD consistently learns token dependence that standard masked diffusion struggles to capture.

### D.2 NEEDLE-IN-A-HAYSTACK TYPE DATA

To directly examine whether VMD can capture strong dependencies between tokens that are far apart, we construct a synthetic dataset that follows a needle-in-a-haystack structure while still allowing exact computation of the KL divergence. Each sequence begins with a four-token needle, ends with an identical needle, and the middle part of the sequence consists entirely of unrelated

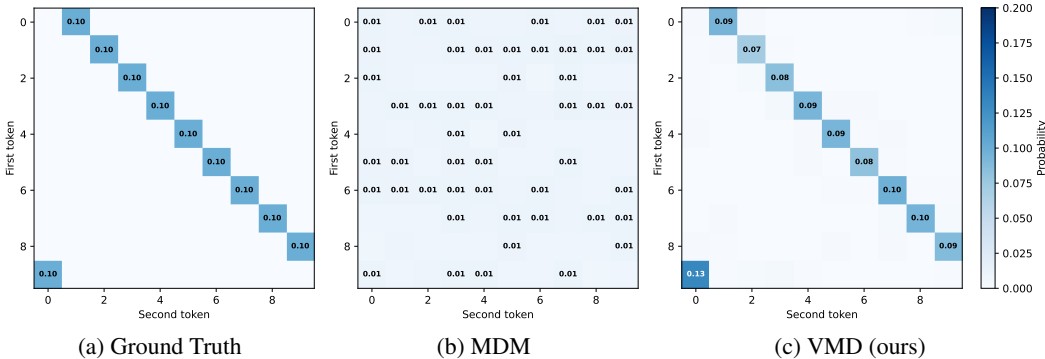

(a) Ground Truth        (b) MDM        (c) VMD (ours)

Figure 4: Results on controlled synthetic data with 2 tokens under the deterministic setting. (a) Ground truth distribution, (b) MDM: baseline masked diffusion with one-step generation, and (c) our VMD with one-step generation. MDM one-step generation is identical to random guessing of each token independently, failing to reflect the true correlations, while VMD closely recovers the ground-truth structure.

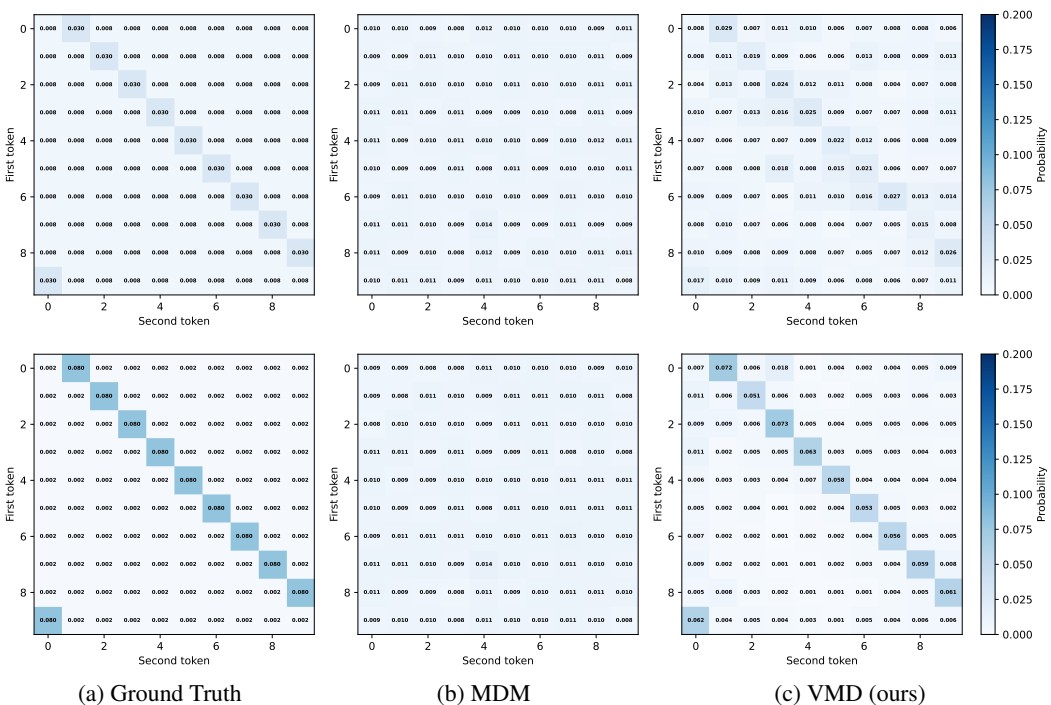

(a) Ground Truth        (b) MDM        (c) VMD (ours)

Figure 5: Results on controlled synthetic data with 2 tokens under the varying correlation setting. (a) Ground truth distribution, (b) MDM: baseline masked diffusion with one-step generation, and (c) our VMD with one-step generation. The first row is for data generated with $p = 0.3$, and the second row is for data generated with $p = 0.8$. The baseline fails to model the data distribution in one-step generation in both cases, producing nearly uniform predictions, while VMD successfully recovers the underlying distributions.

haystack tokens. Formally, we sample a digit $c \sim \text{Unif}\{0, \ldots, 9\}$ and define a four-token needle $n(c) = (c, c + 1, c + 2, c + 3 \mod 10)$. A full sequence of length $L$ is then generated as $x = (n(c), m_1, \ldots, m_{L-8}, n(c))$ where the haystack tokens $m_k$ are drawn i.i.d. also from $\text{Unif}\{0, \ldots, 9\}$. This design creates a controlled long-range dependency between the two needles at the two ends of the sequence.

Table 6: Results on synthetic needle-in-a-haystack data using one-step generation for the entire sequence. VMD captures the dependency between the two needles across distances of at least 30 tokens, while the baseline fails under all tested sequence lengths.

| Method | Metric | $L = 20$ | $L = 30$ | $L = 40$ |
|---|---|---|---|---|
| Baseline | Acc | 0 | 0 | 0 |
| | condAcc | 100 | 100 | 89 |
| | KL | 25.3 | 25.6 | 25.4 |
| VMD | Acc | 90.6 | 87.9 | 4.82 |
| | condAcc | 100 | 100 | 100 |
| | KL | 0.282 | 0.361 | 22.8 |

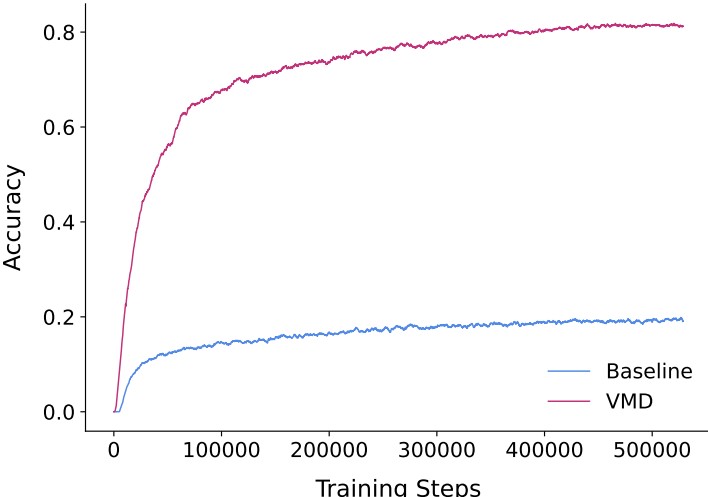

Figure 6: Accuracy (↑) during training on the Sudoku puzzle experiment with NFE=20 under the top probability remasking strategy. VMD consistently outperforms the baseline (MDM) across training iterations. The final accuracy reaches 18.97% for the Baseline and 82.03% for VMD.

We evaluate models using one-step generation for the entire sequence so that the model must reconstruct both needles in a single forward pass. We report three metrics. Accuracy (Acc) measures whether the generated sequence fully matches the needle rule. Conditional accuracy (condAcc) measures whether the model produces a correct sequence when the first token of the first needle is given. KL divergence (KL) measures the gap between the generated distribution and the true distribution of sequences in this dataset.

As shown in Table 6, VMD is able to recover the dependency between the two needles across distances of at least thirty tokens, even with a small model: it achieves high accuracy and low KL divergence under these settings. The masked diffusion baseline struggles on all metrics for all tested sequence lengths. This experiment shows that VMD can capture strong long-range dependencies across multiple tokens and that the restriction to two token blocks in the main paper is a design choice rather than a limitation of the method.

### D.3 SUDOKU DATA

In addition to the final performance reported in the main text, we plot the full training accuracy curves in Figure 6. The figure shows that VMD consistently outperforms the baseline (MDM) throughout training, not only at convergence but also during earlier stages. This confirms that the gains of VMD are stable across the entire training process rather than arising from a specific checkpoint. We also observe that both models converge at a similar rate, indicating that the improvements of VMD come at no additional optimization cost.

Table 7: KL comparison between a fixed latent and a resampled latent on synthetic data. The results show that sampling a new latent at each step does not provide a clear advantage over using a single latent for all steps.

| Method | D1 | | | | D2 | | | |
|---|---|---|---|---|---|---|---|---|
| | $B=2$ NFE$=4$ | $B=2$ NFE$=2$ | $B=1$ NFE$=4$ | $B=1$ NFE$=1$ | $B=2$ NFE$=4$ | $B=2$ NFE$=2$ | $B=1$ NFE$=4$ | $B=1$ NFE$=1$ |
| Baseline | **0.007±0.000** | 2.298±0.189 | **0.010±0.000** | 9.422±0.253 | 0.018±0.001 | 7.656±0.138 | **0.015±0.000** | 7.493±0.211 |
| Fixed $z$ | 0.012±0.008 | **0.045±0.007** | 0.029±0.004 | 0.099±0.007 | **0.017±0.011** | **0.053±0.004** | 0.475±0.007 | **0.093±0.003** |
| Resampled $z$ | 0.008±0.000 | 0.061±0.005 | 0.030±0.003 | **0.089±0.006** | 0.018±0.007 | 0.060±0.005 | 0.552±0.009 | 0.095±0.004 |

Table 8: Accuracy comparison between a fixed latent and a resampled latent on synthetic data. Both variants perform similarly across all settings, confirming that the simpler fixed latent formulation is sufficient in practice.

| Method | D1 | | | | D2 | | | |
|---|---|---|---|---|---|---|---|---|
| | $B=2$ NFE$=4$ | $B=2$ NFE$=2$ | $B=1$ NFE$=4$ | $B=1$ NFE$=1$ | $B=2$ NFE$=4$ | $B=2$ NFE$=2$ | $B=1$ NFE$=4$ | $B=1$ NFE$=1$ |
| Baseline | 99.4±0.04 | 10.1±0.03 | 99.3±0.02 | 0.1±0.00 | 99.3±0.04 | 1.1±0.01 | 99.1±0.03 | 1.1±0.01 |
| Fixed $z$ | **100±0.00** | 97.8±0.01 | **100±0.00** | **93.3±0.01** | **100±0.00** | 96.3±0.01 | **100±0.00** | **94.1±0.00** |
| Resampled $z$ | **100±0.00** | **98.5±0.05** | **100±0.00** | 92.8±0.04 | **100±0.00** | **96.5±0.02** | **100±0.00** | 93.6±0.03 |

# E    ABLATION STUDY

## E.1    TIME DEPENDENT LATENT VARIABLE

In the main text we report results using a time-independent latent variable: all sampling steps share the same latent $z$. This choice follows our model definition and keeps the sampler simple. In principle the latent variable can also be time-dependent. Implementing this variant only requires modifying the sampler so that a new latent $z$ is drawn at every step. We compare these two variants in Tables 7 to 9. The results show that the performance difference between a fixed latent and a resampled latent is very small across all settings. For this reason, we adopt the fixed latent version in the main text for clarity and simplicity of the presentation.

## E.2    BLOCK DEPENDENT LATENT VARIABLE

We further study whether the latent variables should be independent across blocks or conditioned across blocks. The main paper adopts block-independent latent variables because this variant is simpler. In principle, it is possible to introduce block-dependent latent variables by modeling a conditional prior $p_\psi(z^b \mid z^{<b})$. This modification allows information to propagate across blocks through the latent space. However, it also introduces a KL term of the form $D_{\mathrm{KL}}(q_\phi(z^b \mid x_0^{\leq b}, x_t^b)\|p_\psi(z^b \mid z^{<b}))$. This term depends on two moving distributions, which makes optimization more difficult in practice. As shown in Tables 10 and 11, the block dependent latent provides only a mild improvement on synthetic data, and the gain disappears or becomes negative when scaling to text data, where training becomes less stable. For this reason we use the block-independent latent in the main paper while leaving more expressive block-dependent priors as an interesting direction for future work.

## E.3    NOISED DATA DEPENDENT LATENT VARIABLE

In this section we study whether the latent variable should depend on the noised input $x_t$ during inference. The main paper adopts a simple Gaussian prior for $z$. This choice keeps both the sampling procedure and the training objective stable. A natural alternative is to introduce a noised data dependent prior $p_\psi(z \mid x_t)$ and allow the latent variable to adapt to the partially observed input. This modification introduces the KL term $D_{\mathrm{KL}}(q_\phi(z \mid x_0, x_t)\|p_\psi(z \mid x_t))$. Note, this KL divergence involves two moving distributions that must be matched at every timestep, which complicates optimization.

Table 9: Sudoku accuracy when resampling the latent at each time step during inference. The results show that this update does not bring consistent gains over the use of a fixed latent.

| Model | Top prob (Accuracy ↑) | | | Top prob margin (Accuracy ↑) | | |
|---|---|---|---|---|---|---|
| | NFE=5 | NFE=10 | NFE=20 | NFE=5 | NFE=10 | NFE=20 |
| Fixed $z$ | **67.7%** | **76.4%** | 80.9% | **96.9%** | 99.0% | **99.7%** |
| Resampled $z$ | 66.6% | 74.5% | **84.4%** | **96.9%** | **99.4%** | 99.4% |

Table 10: KL comparison between the block-independent and block-dependent latent variant on synthetic data. The block-dependent latent introduces a conditional prior across blocks, yet the improvement over the independent version is mild across all settings.

| Method | D1 | | | | D2 | | | |
|---|---|---|---|---|---|---|---|---|
| | $B=2$ NFE=4 | $B=2$ NFE=2 | $B=1$ NFE=4 | $B=1$ NFE=1 | $B=2$ NFE=4 | $B=2$ NFE=2 | $B=1$ NFE=4 | $B=1$ NFE=1 |
| Baseline | **0.007±0.000** | 2.298±0.189 | **0.010±0.000** | 9.422±0.253 | 0.018±0.001 | 7.656±0.138 | **0.015±0.000** | 7.493±0.211 |
| Block independent $z$ | 0.012±0.008 | 0.045±0.007 | 0.029±0.004 | 0.099±0.007 | **0.017±0.011** | **0.053±0.004** | 0.475±0.007 | **0.093±0.003** |
| Block dependent $z$ | 0.010±0.003 | **0.034±0.001** | 0.023±0.002 | **0.091±0.008** | 0.022±0.001 | 0.064±0.009 | 0.502±0.007 | 0.106±0.008 |

We evaluate this variant on synthetic data as training is relatively well behaved. As reported in Tables 12 and 13, the data-dependent prior does not provide meaningful gains over the fixed Gaussian prior in these controlled settings. When scaling to Sudoku data the training becomes unstable and we observe strong sensitivity to the variance of the learned prior. Stabilizing the model requires additional heuristics such as fixing the variance of $p_\psi$ throughout training. Even with these modifications the final performance remains slightly worse than the simple Gaussian prior baseline.

These findings indicate that using the noised data to condition the latent prior introduces optimization difficulty without bringing consistent benefits. For this reason the main paper adopts the independent Gaussian prior, while more robust designs of noised data-dependent priors are left for future work.

### E.4 UPDATE THE LATENT DURING SAMPLING WITH THE ENCODER

In this ablation we examine whether the encoder can be incorporated into the sampling process by updating the latent variable with the current noised sequence $x_t$ and the predicted denoised sequence $x_0$. This corresponds to replacing the fixed prior sample of $z$ with the approximate posterior $q_\phi(z \mid x_0, x_t)$ at each step of the sampling procedure. The motivation: the encoder can provide additional information about the evolving predictions and could partially mitigate the gap between training and inference.

We evaluate this idea on the Sudoku data using the same VMD as in the main experiments. As shown in Table 14, updating the latent with the encoder produces accuracy comparable to the fixed latent and does not lead to consistent improvements. In several settings the performance is slightly worse, suggesting that repeatedly updating $z$ introduces instability without providing a clear benefit. These findings indicate that the fixed latent formulation used in the main text is already a strong and stable choice. More systematic exploration of encoder based updates may still be valuable and we leave this direction for future work.

### E.5 FINE-TUNE FROM PRETRAINED MDM

In this section we study whether initializing VMD from a pretrained masked diffusion model can improve performance. We compare three variants: training VMD from scratch, initializing the decoder from a pretrained MDM, and initializing from a pretrained MDM while freezing all loaded weights. The results are summarized in Tables 15 and 16. We observe that freezing the pretrained model consistently leads to worse KL divergence and accuracy, indicating that the pretrained representation is not directly compatible with the latent conditioned formulation of VMD. Unfreezing the pretrained decoder improves stability but only recovers performance close to training VMD from scratch. This suggests that the latent variable fundamentally changes the inductive bias of the model and that simple weight transfer from an unconditional MDM does not provide a meaningful advan-

Table 11: Accuracy comparison between the block-independent and block-dependent latent variant on synthetic data. The block-dependent design provides no gains, while the block-independent version remains competitive and more stable in practice.

| Method | D1 | | | | D2 | | | |
|---|---|---|---|---|---|---|---|---|
| | $B=2$ NFE=4 | $B=2$ NFE=2 | $B=1$ NFE=4 | $B=1$ NFE=1 | $B=2$ NFE=4 | $B=2$ NFE=2 | $B=1$ NFE=4 | $B=1$ NFE=1 |
| Baseline | 99.4±0.04 | 10.1±0.03 | 99.3±0.02 | 0.1±0.00 | 99.3±0.04 | 1.1±0.01 | 99.1±0.03 | 1.1±0.01 |
| Block independent $z$ | **100±0.00** | **97.8±0.01** | **100±0.00** | 93.3±0.01 | **100±0.00** | **96.3±0.01** | **100±0.00** | **94.1±0.00** |
| Block dependent $z$ | **100±0.00** | 97.7±0.03 | **100±0.00** | **94.7±0.02** | **100±0.00** | 96.1±0.01 | **100±0.00** | 93.9±0.03 |

Table 12: KL comparison between the vanilla version and a noised data-dependent latent on synthetic data. The improvement over the vanilla version is mild across all settings.

| Method | D1 | | | | D2 | | | |
|---|---|---|---|---|---|---|---|---|
| | $B=2$ NFE=4 | $B=2$ NFE=2 | $B=1$ NFE=4 | $B=1$ NFE=1 | $B=2$ NFE=4 | $B=2$ NFE=2 | $B=1$ NFE=4 | $B=1$ NFE=1 |
| Baseline | **0.007±0.000** | 2.298±0.189 | 0.010±0.000 | 9.422±0.253 | 0.018±0.001 | 7.656±0.138 | 0.015±0.000 | 7.493±0.211 |
| $p(z)$ | 0.012±0.008 | 0.045±0.007 | 0.029±0.004 | 0.099±0.007 | **0.017±0.011** | **0.053±0.004** | 0.475±0.007 | **0.093±0.003** |
| $p_\psi(z \mid x_t)$ | 0.011±0.005 | **0.036±0.003** | **0.004±0.000** | **0.056±0.002** | 0.031±0.002 | 0.063±0.004 | **0.015±0.002** | 0.108±0.004 |

tage. Larger backbones may benefit more from pretraining, so exploring this direction remains an interesting direction for future work.

# F  LLM USAGE

While preparing this work, we used a large language model (LLM) to assist with language editing. The LLM's contributions were limited to improving the clarity of the text. The core research, experimental design, and all scientific claims remain our original work.

Table 13: Accuracy comparison between the vanilla version and a noised data-dependent latent on synthetic data. The improvement over the vanilla version is mild across all settings.

| Method | D1 | | | | D2 | | | |
|---|---|---|---|---|---|---|---|---|
| | $B=2$ NFE$=4$ | $B=2$ NFE$=2$ | $B=1$ NFE$=4$ | $B=1$ NFE$=1$ | $B=2$ NFE$=4$ | $B=2$ NFE$=2$ | $B=1$ NFE$=4$ | $B=1$ NFE$=1$ |
| Baseline | 99.4±0.04 | 10.1±0.03 | 99.3±0.02 | 0.1±0.00 | 99.3±0.04 | 1.1±0.01 | 99.1±0.03 | 1.1±0.01 |
| $p(z)$ | **100±0.00** | **97.8±0.01** | **100±0.00** | 93.3±0.01 | **100±0.00** | 96.3±0.01 | **100±0.00** | **94.1±0.00** |
| $p_\psi(z \mid x_t)$ | **100±0.00** | 97.6±0.02 | **100±0.00** | **94.7±0.00** | **100±0.00** | **96.9±0.02** | **100±0.00** | 92.6±0.02 |

Table 14: Sudoku accuracy when updating the latent during sampling using the encoder $q_\phi(z \mid \hat{x}_0, x_t)$. The results show that this update does not bring consistent gains over the use of a fixed latent.

| Model | Top prob (Accuracy ↑) | | | Top prob margin (Accuracy ↑) | | |
|---|---|---|---|---|---|---|
| | NFE=5 | NFE=10 | NFE=20 | NFE=5 | NFE=10 | NFE=20 |
| VMD | **67.7%** | **76.4%** | **80.9%** | **96.9%** | 99.0% | **99.7%** |
| VMD w/ $q_\phi$ | 66.8% | 74.4% | 79.6% | **96.9%** | **99.3%** | 99.3% |

Table 15: KL comparison between VMD, VMD initialized from a pretrained MDM, and VMD using pretrained MDM with frozen weights. The results show that freezing the pretrained model leads to worse performance and that unfreezing it only recovers results close to training VMD from scratch.

| Method | D1 | | | | D2 | | | |
|---|---|---|---|---|---|---|---|---|
| | $B=2$ NFE$=4$ | $B=2$ NFE$=2$ | $B=1$ NFE$=4$ | $B=1$ NFE$=1$ | $B=2$ NFE$=4$ | $B=2$ NFE$=2$ | $B=1$ NFE$=4$ | $B=1$ NFE$=1$ |
| Baseline | **0.007±0.000** | 2.298±0.189 | **0.010±0.000** | 9.422±0.253 | 0.018±0.001 | 7.656±0.138 | **0.015±0.000** | 7.493±0.211 |
| VMD | 0.012±0.008 | **0.045±0.007** | 0.029±0.004 | **0.099±0.007** | **0.017±0.011** | **0.053±0.004** | 0.475±0.007 | **0.093±0.003** |
| Pretrained | 0.005±0.001 | **0.045±0.003** | 0.027±0.005 | 0.103±0.005 | **0.017±0.000** | 0.064±0.007 | 0.123±0.004 | 0.107±0.009 |
| Pretrained Frozen | 0.002±0.001 | 0.057±0.001 | 0.012±0.001 | 0.111±0.004 | 0.029±0.002 | 0.125±0.002 | 0.034±0.002 | 0.120±0.002 |

Table 16: Accuracy comparison between VMD, VMD initialized from a pretrained MDM, and VMD using pretrained MDM with frozen weights. The results show that freezing the pretrained model leads to worse accuracy and that unfreezing it only recovers results close to training VMD from scratch.

| Method | D1 | | | | D2 | | | |
|---|---|---|---|---|---|---|---|---|
| | $B=2$ NFE$=4$ | $B=2$ NFE$=2$ | $B=1$ NFE$=4$ | $B=1$ NFE$=1$ | $B=2$ NFE$=4$ | $B=2$ NFE$=2$ | $B=1$ NFE$=4$ | $B=1$ NFE$=1$ |
| Baseline | 99.4±0.04 | 10.1±0.03 | 99.3±0.02 | 0.1±0.00 | 99.3±0.04 | 1.1±0.01 | 99.1±0.03 | 1.1±0.01 |
| VMD | **100±0.00** | **97.8±0.01** | **100±0.00** | **93.3±0.01** | **100±0.00** | **96.3±0.01** | **100±0.00** | **94.1±0.00** |
| Pretrained | **100±0.00** | 97.3±0.21 | **100±0.00** | **93.3±0.02** | **100±0.00** | 95.3±0.11 | 99.5±0.04 | 92.6±0.31 |
| Pretrained Frozen | **100±0.00** | 96.7±0.11 | **100±0.00** | 90.5±0.29 | 99.8±0.03 | 91.1±0.11 | 99.1±0.05 | 90.9±0.08 |

