# OpenReview forum: "Variational Masked Diffusion Models"
_ICLR.cc/2026/Conference — Submitted to ICLR 2026_

### Official Review · Reviewer_MC7Q · 2025-10-19

**Soundness:** 2
**Presentation:** 3
**Contribution:** 2
**Rating:** 4
**Confidence:** 4

**Summary:**

The paper proposes Variational Masked Diffusion (VMD), a framework designed to address the limitation of independent concurrent token generation in standard masked diffusion models. VMD introduces a global latent variable to model joint dependencies among concurrently predicted tokens. Training of VMD is performed via variational inference over this latent variable. The inference of VMD starts by first sampling the latent variable and then iteratively unmasking tokens conditioned on the sampled latent variable. The framework is further extended to block diffusion models, combining autoregressive dependencies across blocks with variational diffusion within blocks. The proposed approach is evaluated on synthetic datasets, Sudoku puzzles, and the Text8 dataset.

**Strengths:**

- The paper studies a key problem of discrete diffusion models in capturing the joint dependencies of concurrently predicted tokens.
- The idea of using a latent variable to capture the joint dependencies is novel and has not been studied in the discrete diffusion literature (as far as I am aware).
- The authors have provided reasonably comprehensive experiments on synthetic data, but it would be interesting to get more comprehensive experiments on text.

**Weaknesses:**

- The basic formulation of the latent variable in Section 3 has an underlying assumption that z is independent of x_t. More concretely, in eq.(3) and eq.(4), there should be p(z / x_t) instead of p(z). I believe this approximation may lead to not enough improvement when the distance between p(z / x_t) and p(z) is large. Evidence of this problem appears to be in the results on text data in Section 4.4, where the improvement of VMD over BD3-LM for block size 8 is much smaller than for block size 4. This might be true because for x_t^b of block size 8 can have more information and hence p(z / x_t) can be different from p(z).

- While the experiments are quite comprehensive for synthetic data, they are somewhat lacking for text. It would be interesting to see the metrics like MAUVE and generative perplexity when using VMD with different numbers of inference steps. I wonder if, for a small number of inference steps, VMD provides a larger benefit compared to BD3-LM or MDM in general. This is possible because the model is required to more accurately capture the joint dependencies of concurrently predicted tokens.

**Questions:**

The training of VMD uses q_{\phi}(z | x_0, x_t) - an approximate posterior parameterized by trainable parameters ϕ, but the inference doesn’t. Is there a possibility of using it to partially solve the issue mentioned in the first weakness? x_0 can be approximated with the denoised sequence before remasking during inference.

---

> ### Author Response · Authors · 2025-11-22
> **Response to Reviewer MC7Q**
>
> Thanks for your time and for highlighting the importance of the problem and the novelty of using a latent variable to capture joint dependencies in discrete diffusion models.
>
> ***QF1: The formulation in Sec. 3 assumes that $z$ is independent of $x_t$. In Eqs. (3) and (4), there should be $p(z | x_t)$ instead of $p(z)$. I believe this approximation may limit improvement. Evidence of this appears to be in the results on text data in Sec. 4.4: the improvement of VMD over BD3-LM for block size 8 is much smaller than for block size 4. This might be true because for $x_t^b$ of block size 8 can have more information and hence $p(z | x_t)$ can be different from $p(z)$.***
>
> > Our goal is to demonstrate feasibility of VMD, so we adopt the simplest setting: latent does not depend on noised data $x_t$. It is indeed possible to use $x_t$ to obtain $z$ by introducing a prior $p_\psi(z \mid x_t)$. This creates a KL term of the form $D_\text{KL}(q_{\phi}(z \mid x_0, x_t) \| p_\psi(z \mid x_t))$ which involves two moving distributions. This coupling makes optimization more involved. We tested this on synthetic data and observed no significant improvement (see below). When scaled to Sudoku data, training became unstable. When fixing the variance of $p_\psi$ to stabilize training, results were slightly worse. Details are shown in Appendix E.3. We view this noised data dependent prior as a great direction for future work.
>
> D1 ContPairs4
>
> | KL | B=2 NFE=4 | B=2 NFE=2 | B=1 NFE=4 | B=1 NFE=1 |
> |--|--|--|--|--|
> | Baseline | 0.007±0.000 | 2.298±0.189 | 0.010±0.000 | 9.422±0.253 |
> | $p(z)$ | 0.012±0.008 | 0.045±0.007 | 0.029±0.004 | 0.099±0.007 |
> | $p_\psi(z\|x_t)$ | 0.011±0.005 | 0.036±0.003 | 0.004±0.000 | 0.056±0.002 |
>
> | Accuracy | B=2 NFE=4 | B=2 NFE=2 | B=1 NFE=4 | B=1 NFE=1 |
> |--|--|--|--|--|
> | Baseline | 99.4±0.04 | 10.1±0.03 | 99.3±0.02 | 0.1±0.00 |
> | $p(z)$ | 100±0.00 | 97.8±0.01 | 100±0.00 | 93.3±0.01 |
> | $p_\psi(z\|x_t)$ | 100±0.00 | 97.6±0.02 | 100±0.00 | 94.7±0.00 |
>
> D2 RandPairs4
>
> | KL | B=2 NFE=4 | B=2 NFE=2 | B=1 NFE=4 | B=1 NFE=1 |
> |--|--|--|--|--|
> | Baseline | 0.018±0.001 | 7.656±0.138 | 0.015±0.000 | 7.493±0.211 |
> | $p(z)$ | 0.017±0.011 | 0.053±0.004 | 0.475±0.007 | 0.093±0.003 |
> | $p_\psi(z\|x_t)$ | 0.031±0.002 | 0.063±0.004 | 0.015±0.002 | 0.108±0.004 |
>
> | Accuracy | B=2 NFE=4 | B=2 NFE=2 | B=1 NFE=4 | B=1 NFE=1 |
> |--|--|--|--|--|
> | Baseline | 99.3±0.04 | 1.1±0.01 | 99.1±0.03 | 1.1±0.01 |
> | $p(z)$ | 100±0.00 | 96.3±0.01 | 100±0.00 | 94.1±0.00 |
> | $p_\psi(z\|x_t)$ | 100±0.00 | 96.9±0.02 | 100±0.00 | 92.6±0.02 |
>
>
> ***QF2: Experiments are comprehensive for synthetic data, but somewhat lacking for text. Show metrics like MAUVE and generative perplexity when using VMD with different numbers of inference steps. For few inference steps, VMD may provide a larger benefit compared to BD3-LM or MDM in general?***
>
> > Thank you for the suggestion. We are currently running larger scale text experiments on a DiT backbone and will report generative perplexity and MAUVE across different inference steps once training finishes. We expect these results to clarify the benefit of VMD at low NFE and answer your question.
>
> ***QF3: Training of VMD uses $q_{\phi}(z | x_0, x_t)$ - an approximate posterior parameterized by trainable parameters $\phi$, but inference doesn’t. Is it possible to use it? x_0 can be approximated with the denoised sequence before remasking during inference.***
>
> > Our goal is to demonstrate feasibility of VMD, so we adopt the simplest setting: sampling does not depend on the encoder. It is indeed possible to use $q_\phi$ to update $z$ with current $x_t$ and predicted $x_0$. We tested this on Sudoku data and observed no significant improvement (see below). Details are shown in Appendix E.4. Due to time constraints we did not pursue this line of investigation further, but we agree that a more systematic study of incorporating the encoder into the sampling is a promising future direction.
>
> | Accuracy |  |  Top prob  |  | Top prob  | margin    |   |
> |--------------|:------------:|:----:|:----:|:-------------:|:----:|:---:|
> |              | NFE=5                    | NFE=10 | NFE=20 | NFE=5 | NFE=10 | NFE=20 |
> | VMD      | 67.7%               | 76.4% | 80.9% | 96.9% | 99.0% | 99.7% |
> | VMD w/ $q_\phi$ | 66.8%                     | 74.4%  | 79.6%  | 96.9% | 99.3% | 99.3% |

---

### Official Review · Reviewer_GQc8 · 2025-10-21

**Soundness:** 4
**Presentation:** 3
**Contribution:** 2
**Rating:** 4
**Confidence:** 4

**Summary:**

Discrete diffusion models have can flip multiple tokens at once, from the product distribution of their marginals, leading to poor modeling of inter token dependencies. To address this issue, the authors introduce latent variables such that the model is conditionally independent given the latents. This formulation is conceptually similar to the encoder-decoder setting used in Variational AutoEncoders and is trained with a suitable ELBO loss. The authors give a similar extension to the Block Diffusion Framework.

The efficacy of the framework is demonstrated with synthetic experiments, SuDoKu and language modeling with the text8 dataset.

**Strengths:**

* The setting considered by the work is very important, especially to improve the performance of diffusion models. I believe that the presented ideas are in the right direction.

* The idea is cleanly formulated and the evaluations on synthetic datasets demonstrates its efficacy.

* The method shows some gains in solving SuDoKu problems, especially at low NFE.

**Weaknesses:**

* The training procedure comes with additional complexities, where an additional encoder needs to be trained. Training and tuning this correctly can be an additional burden.

* The experiments on text data show only marginal gains -- which I am not even sure is statistically significant and worth the effort of training an additional latent model.

* The gains in the SuDoKu experiments are marginal and the presentation is not very clear. It would be helpful if $c_{prob}$ and $c_{marg}$ are recalled and explained in this section more rigorously (also see questions).

**Questions:**

* The authors use the pipeline from Kim et al (2025), along with their method as the baseline. However, I could not find the SuDoKu results for every NFE value presented in this reference since it only considers NFE=50. Please point me to the correct numbers in case I missed it. This is important since the authors claim that they reproduce these values in Table 4.

---

> ### Author Response · Authors · 2025-11-22
> **Response to Reviewer GQc8**
>
> Thanks for your time and for highlighting the importance of the setting we study and for noting that the proposed ideas move in a promising direction.
>
> ***QE1: Training comes with additional complexities (additional encoder needs to be trained). Training and tuning this correctly can be an additional burden.***
>
> > Yes, training uses an additional encoder, which adds some complexity and tuning cost. However, this overhead is worthwhile for data with strong structural dependencies such as Sudoku where MDM struggles to capture the joint distribution and fails to do low NFE generation.
>
> ***QE2: Experiments on text data show marginal gains.***
>
> > On text8 we do observe a consistent improvement in test perplexity, showing that VMD produces higher quality samples than MDM. We are currently running VMD on a DiT backbone comparable in size to GPT-2 small to provide a more comprehensive evaluation of token dependency on larger text data. We will update the paper once these experiments complete and we hope this addresses your concern.
>
> ***QE3: Gains on Sudoku are marginal and presentation is not clear. It would be helpful if $c_{prob}$ and $c_{marg}$ are recalled and explained in this section more rigorously.***
>
> > We revised the architecture. Originally, we added the latent at the input layer, which isn’t very expressive. We now use an adaptive layer normalization module. This change enables the latent to more effectively interact with the data. Details are in Appendix C with updates marked in blue.
>
> > The new architecture leads to a large improvement. As shown in Tab. 3 and Fig. 6, with top probability sampling MDM reaches 20.4% accuracy while VMD reaches 80.9%. The latent now provides a stronger signal and significantly improves Sudoku results.
>
> > As requested, we recall and explain both $c_{prob}$ and $c_{marg}$ in Sec. 4.3 of the main text. The updated definitions clarify how each sampling strategy is computed and how they are used in our experiments.
>
> ***QE4: For baseline from Kim et al (2025) I could not find the Sudoku results for reported NFE. Please point me to the correct numbers in case I missed it. This is important since the authors claim that they reproduce these values in Tab. 3.***
>
> > Kim et al. (2025) report Sudoku results only at NFE=50, which are 18.51% for top probability and 89.49% for top probability margin. Reproducing results, we found that the same level of accuracy is already reached at NFE=20, with 20.4% and 91.1%. For this reason, Tab. 3 reports the NFE=20 results and states that we reproduced their results. To highlight the advantage of VMD at lower NFE, we additionally include experiments at NFE=5 and NFE=10.

---

> ### Comment · Reviewer_GQc8 · 2025-11-28
> **Response from the reviewer**
>
> Thank you for the clarifications. I am happy with the SuDoKu results. I disagree that the text8 data shows consistent improvement in perplexity (2.873 vs 2.858). This could very much be within the margin of statistical error. I would love to see GPT-2 results once updated.
>
> Even if GPT-2 results turn out to be great, like the SuDoKu results, I am not happy with the paper including the 2 of their best results (along with architectural modifications) during the rebuttal phase rather than during the submission itself. I will retain my score and suggest that the authors submit it to another venue.
>
> I want to reiterate that I like this approach and the problem statement. With better execution, this contribution will be valued by the community.

---

> > ### Author Response · Authors · 2025-11-28
> >
> > ***QE5: Even if GPT-2 results turn out to be great, like the SuDoKu results, I am not happy with the paper including the 2 of their best results (along with architectural modifications) during the rebuttal phase rather than during the submission itself. I will retain my score and suggest that the authors submit it to another venue.***
> >
> > > Thank you for sharing this comment. Our understanding of the ICLR review process: authors are encouraged to improve the submission and to report additional results when reviewers request clarifications or further experiments. Many ICLR papers include updated experiments during the rebuttal stage for this reason. We did not introduce unrelated changes. Instead, we provided additional evaluations that reviewers asked for, using exactly the method described in the submission. We would like to note that we neither presented an architecture choice as a contribution, nor did we describe it as part of the method section. We hope the reviewer assessment can focus on the scientific contribution of our method in the current revision.

---

### Official Review · Reviewer_dRUz · 2025-10-26

**Soundness:** 3
**Presentation:** 3
**Contribution:** 2
**Rating:** 4
**Confidence:** 4

**Summary:**

This paper proposes variational masked diffusion (VMD) that introduces a latent variable to masked diffusion models in order to capture the dependencies among the predicted positions given a partially masked sequence during multi-token generation. The training loss is derived from the ELBO of the data likelihood, similar to VAEs. The authors also propose a blockwise formulation that augments the block masked diffusion model. The experiments demonstrate that VMD outperforms existing model structures on various modeling tasks on synthetic data and language modeling.

**Strengths:**

The paper is well-written and easy to follow, with clean mathematical derivations. The experimental results contain diverse tasks, which demonstrate the effectiveness of the proposed method in a convincing manner.

The main contribution of this paper is to introduce a latent variable into masked diffusion models, and train it in a VAE-like manner with a small overhead of the encoder and decoder networks. This is an interesting idea that has not been widely explored in the masked diffusion literature, but I'd like to mention that a prior work **VADD (arXiv:2505.17384)** has already explored similar concepts, and the only novelty of this paper compared with VADD seems to be the blockwise formulation, which is a relatively minor extension. While I acknowledge that according to the review policy this is not grounds for rejection, I strongly encourage the authors to discuss this prior work and clearly articulate the differences and contributions of your paper.

**Weaknesses:**

As mentioned above, the authors are suggested to incorporate a more comprehensive literature review of related works.

I appreciate the authors for providing experimental results on various modalities. However, these are all relatively small-scale:

- The paper takes at least two pages to present the experiments on synthetic data with only 2 or 4 dimensions, which seems too trivial. In these tasks, the data distribution has strong correlations among dimensions, so it is expected that introducing a latent variable would help compared with independent decoding. Therefore, this serves more as a sanity check rather than a convincing demonstration of the effectiveness of the proposed method in real-world scenarios.

- For the experiments on Sudoku, while it is a more complex task, the improvement over the vanilla masked diffusion model is relatively marginal, and surprisingly, the improvement at NFE = 5 ($+1.3\%$) is much smaller than that at NFE = 10 and 20 ($+2.3\%$ and $+2.5\%$), which is a little bit counter-intuitive to me since I expect the advantage of modeling dependencies should be more pronounced when the NFE is smaller.

- Finally, for language modeling, the dataset used is text8, which is quite small and outdated compared with more recent larger-scale datasets used in discrete diffusion models, which are at least on GPT-2 level. Thus, it remains unclear whether the proposed method can scale to more complex and larger-scale language modeling tasks. The improvement of VMD over BD3-LM is also relatively insignificant, which may also be due to the small scale of the dataset.

**Questions:**

1. In section 3, how to go from (3) to (4)? It seems that a better way to present this is to define $p _ \theta(x _ s|x _ t)$ by $\int \prod _ {i=1}^d p _ \theta(x _ s^i|x _ t,z)\cdot p(z)\mathrm{d}z$ first, which directly leads to (3) and (4) by partially integrating out dimensions in $x _ s$.

2. Could the authors provide a demonstration of how to modify the network architectures for masked diffusion models in order to receive the latent variable as input? A more interesting question is, can we use a pretrained masked diffusion model to adapt to VMD by freezing (or training with small learning rate) the original network weights and only training the overhead? Experiments along this direction would be interesting.

3. For table 2, it is better to convert it to a plot with $p$ on the x-axis and KL on the y-axis, so that we can better visualize the trend.

4. I'm thinking about the design of the latent variable $z$, which is trained to capture the dependencies among different masked dimensions in a partially observed sequence. Setting the prior of $z$ and the output of encoder as Gaussian distributions is of course a standard choice, but I feel that it may not be optimal in this case due to the lack of semantic meaning. Do the authors have any thoughts on this? For example, would it be better to use a more complex prior such as a mixture of Gaussians, or even a discrete latent variable? Moreover, for sampling (algorithm 2), I think it is also possible to resample the latent variable $z$ at each step instead of only once at the beginning. Have the authors considered this? Would it improve the performance?

---

> ### Author Response · Authors · 2025-11-22
> **Response to Reviewer dRUz**
>
> Thanks for your time and for the positive feedback on writing, clarity, and experimental diversity.
>
> ***QD1: Incorporate a more comprehensive literature review.***
>
> > Thanks for pointing out VADD. We apologize for not having been aware of it. VMD and VADD were developed concurrently and independently. In the revised paper, we cited and discussed VADD in the related work: Although both methods use a similar training objective, VMD introduces a variational latent variable together with a block structure. This results in more control over the range and structure of the latent representation. Our experiments, including synthetic tasks that explicitly control token dependence, are designed to demonstrate these effects. This isn’t discussed in VADD.
>
> ***QD2: Experiments on synthetic data with only 2 or 4 dimensions seems too trivial.***
>
> > We believe low-dimensional experiments are essential. They provide a clear and controllable setting to isolate and verify key behavior of our method, and they clearly highlight the issue of parallel decoding in masked diffusion models when tokens are dependent. For the 2 token case, we can visualize the distribution as shown in Figs. 3 to 5. For the 4 token case, we highlight the importance of the block structure and show how it enables the model to capture multi-token dependencies. These controlled results provide the foundation to interpret gains observed on real data.
>
> ***QD3: For Sudoku, the improvement over masked diffusion is relatively marginal. Surprisingly, improvement at low NFE is smaller.***
>
> > Great observation. We also noticed this and revised the architecture. Originally, we added the latent at the input layer, which isn’t very expressive. We now use an adaptive layer normalization module. This change enables the latent to more effectively interact with the data. Details are in Appendix C with updates marked in blue.
>
> > The new architecture leads to a large improvement. As shown in Tab. 3 and Fig. 6, with top probability sampling MDM reaches 20.4% accuracy while VMD reaches 80.9%. The latent now provides a stronger signal and significantly improves Sudoku results.
>
> ***QD4: Text8 is quite small; models are at least on GPT-2 level. Unclear scalability? The improvement of VMD over BD3-LM is small.***
>
> > Thank you for the suggestion. Due to GPU limitations we were not able to run large scale text experiments in the initial submission. We are currently running VMD on a DiT backbone with a size comparable to GPT-2 small and expect to obtain results within the coming week. We will update the paper once these experiments finish and we agree that larger backbone evaluations will further demonstrate the effectiveness of VMD.
>
> ***QD5: How to go from Eq. (3) to (4)?***
>
> > Great suggestion. We revised the text as recommended.
>
> ***QD6: Explain how to adopt architectures to receive the latent? Adapt a pretrained masked diffusion model to VMD?***
>
> > Great questions. Architecture modifications for VMD are described in Appendix C: latent is injected into masked diffusion via adaptive layernorm.
>
> > Adapting a pretrained masked diffusion model is an interesting direction. We tested this on synthetic datasets and found that freezing the pretrained model leads to worse results and that unfreezing it only recovers results close to training VMD from scratch. The details are shown in Appendix E.5. Due to time constraints we did not pursue this line of investigation further, but we agree that a more systematic study of pretrained adaptation is a promising future direction.
>
>
> ***QD7: Plot instead of Tab. 2.***
>
> > Great suggestion. We replaced Tab. 2 with a plot.
>
> ***QD8: Using a Gaussian prior may not be optimal. Is a more complex prior better?***
>
> > Our goal is to demonstrate feasibility of VMD, so we adopt the simplest setting: time-independent and block-independent latent. It is indeed possible to use a more complex prior. We include an ablation study with different latent formulations in Appendix E: resampling latent $z_t$ at each time step, block-dependent latent variables $p_\psi(z^b|z^{<b})$, noised data-dependent latent variables $p_\psi(z|x_t)$, and updating the latent during sampling with the encoder $q_\phi(z|x_0,x_t)$. These results show: while more complex priors are feasible, the simple Gaussian prior already provides stable training and strong performance. A full exploration of latent distributions is an interesting direction for future work.
>
> ***QD9: Resample the latent variable $z$ at each step instead of only once at the beginning.***
>
> > Resampling the latent $z$ at each step is discussed in Appendix E.1. It leads to no significant improvement. We therefore adopt a fixed latent $z$ in the main paper, and we view use of other priors as an interesting direction for future work.

---

### Official Review · Reviewer_8mJQ · 2025-10-31

**Soundness:** 4
**Presentation:** 2
**Contribution:** 2
**Rating:** 4
**Confidence:** 5

**Summary:**

This paper proposes VDM that adds a global latent variable z to MDM so that the distribution is jointly conditioned on a partially masked sequence, and the latent variable reduces the token dependency. Addressing token dependency is crucial for MDM in the sense that its parallel sampling nature can lead to sampling errors since the MDM itself doesn't model the token dependency. This work provides (synthetic) experimental results to support their claim on the token dependency.

**Strengths:**

The idea of tacking token dependency through a latent variable seems correct, as it provides more information on the posterior distribution that cannot be given solely by the partially masked sequence. Also, the ELBO object is theoretically grounded, resulting in a reasonable training loss. Moreover, the experiments are well-designed to show that the VDM indeed captures the token dependency much better.

**Weaknesses:**

The experimental claim in this paper is apparently weak.
1. In the synthetic dataset, the experiment is well-designed and
2. In the text data, although I appreciate the author's effort on pretraining VDM from scratch, the small difference in generative perplexity (Table 5) isn't enough to tell that VDM is much better than MDM in capturing the token dependency.
3. In the Sudoku puzzle, the VDM's accuracy is also marginally better than baseline, e.g., Top Prob margin. Given that it's a small-scale experiment (not done with large-scale MDM), this small difference in accuracy cannot firmly support the claim. Moreover, I don't fully get the reason why VDM would work better than MDM in the Sudoku setup, where the answer to a given puzzle is often deterministic. This is because even though we provide z as an additional conditional variable, if the answer is fixed given the partially filled board, then the latent variable z isn't playing any meaningful role, i.e, p_true (x_0^i | x_t) = p_true (x_0^i | x_t ,z). I believe this was probably the fundamental reason why VDM failed to outperform MDM baselines.

Given this insight, I believe there will certainly be an experiment setup where (1) the answer is not uniquely defined, so that providing z as a context could be meaningful, (2) modeling the token dependence is crucial to get a good result, i.e., logic puzzles. I understand that the main contribution could be interpreted as formulating the VLM; however, I believe ideally the authors could've shown a successful case in which VDM outperforms MDM, further enhancing its applicability.

**Questions:**

I wonder how the authors think about my point in the Weakness section!

---

> ### Author Response · Authors · 2025-11-22
> **Response to Reviewer 8mJQ**
>
> ***QC1: For text data, generative perplexity (Table 4) isn't enough to tell that VMD is much better than MDM in capturing the token dependency.***
>
> > On text8 we do observe a consistent improvement in test perplexity, showing that VMD produces higher quality samples than MDM. We are currently running VMD on a DiT backbone comparable in size to GPT-2 small to provide a more comprehensive evaluation of token dependency on larger text data. We will update the paper once these experiments complete and we hope this addresses your concern.
>
> ***QC2: For Sudoku, VMD's accuracy is marginally better than baseline. Reason why VDM would work better than MDM for Sudoku data, as answer is often deterministic.***
>
> > We revised the architecture to better consider the latent variable. Originally, we added the latent at the input layer. We now use an adaptive layer normalization module. This change enables the latent to more effectively interact with the data. Details are in Appendix C with updates marked in blue.
>
> > The new architecture leads to a large improvement. As shown in Tab. 3 and Fig. 6, with top probability sampling MDM reaches 20.4% accuracy while VMD reaches 80.9%. The latent now provides a stronger signal and significantly improves Sudoku results.
>
> > Regarding benefits of VDM on Sudoku: the situation is in fact the opposite of what the reviewer suggested. A unique solution creates very strong long range dependencies which is exactly where the latent is most useful. As highlighted in the introduction using the poker example and in our synthetic experiments, the more deterministic the structure, the more helpful the latent. MDM cannot capture this dependence, while VMD uses the latent to coordinate the entire solution. This explains its much higher accuracy for Sudoku data.

---

### Official Review · Reviewer_yBdK · 2025-11-01

**Soundness:** 3
**Presentation:** 4
**Contribution:** 4
**Rating:** 6
**Confidence:** 3

**Summary:**

This work considers the problem of dependence between sampled tokens from masked diffusion models (MDMs). One-step sampling in MDMs does not consider dependence between tokens, while multi-step sampling is significantly slower. This work proposes Variational Masked Diffusion Models (VMDs) that predict an intermediate latent variable that, when conditioned on, renders individual tokens independent of each other.

**Strengths:**

The paper studies an interesting problem and presents it well with a nice exposition. The topic is also important as it might reduce inference time while maintaining quality outputs. I think this work, either by itself or through its follow-ups, can impact real-world large language models.

**Weaknesses:**

I did not find major weaknesses in this work. Some minor questions are given below. Additional minor questions/comments on writing are provided under "questions."

**W1. Number of tokens per block**: Since VMDs have key advantages on sequences with high inter-token dependency, why are the experiments limited to at most 2 tokens per block? Can VMDs work with multiple tokens with strong dependencies between tokens that are located far from each other? Maybe on a needle-in-a-haystack type of dataset [A1], even if it is a synthetic one that is quicker to train on. I know the Sudoku experiment has more than two tokens per block, but you cannot measure the KL divergence in that experiment.

**W2. Obtaining inference-time latent $z$**: In Algorithm 2, the input is a fully or partially masked input sequence $x$. But $z$ is simply sampled from a Gaussian. When a partially masked input sequence is available, is it used to obtain $z$? If not, why?

**W3. Effect of resampling $z$**: Related to the previous question, does sampling different $z$ for the same starting sequence result in different outputs?

**Questions:**

These are some minor writing-related comments/questions:

**Q1.** "NFE" used in Table 3 is defined in the appendix, but not in the main text.

**References**

[A1] Elliot Nelson, Georgios Kollias, Payel Das, Subhajit Chaudhury, Soham Dan, "Needle in the Haystack for Memory Based Large Language Models", ArXiv 2024.

---

> ### Author Response · Authors · 2025-11-22
> **Response to Reviewer yBdK**
>
> Thanks for your time and for highlighting that the problem we study is interesting and well presented. We appreciate your comments on the potential impact of this work on reducing inference time.
>
> ***QB1: Why are experiments limited to at most 2 tokens per block? Can VMDs work with multiple tokens with strong dependencies between tokens that are located far from each other? Maybe on needle-in-a-haystack type data. Sudoku data has more than two tokens per block, but you cannot measure KL divergence.***
>
> > To examine whether VMD captures dependencies between tokens that are far apart while allowing KL computation, we construct synthetic needle-in-a-haystack data (see Appendix D.2 for details): Each sequence with length $L$ begins with a four-token needle, ends with an identical needle, and the middle part consists of unrelated haystack tokens. We evaluate using one-step generation, i.e., the model must reconstruct both needles in a single forward pass. We report accuracy, conditional accuracy (whether model produces a correct sequence when the first token of the first needle is given), KL divergence. As shown below, VMD recovers the two needles across distances of at least thirty tokens, even with a small model. The masked diffusion baseline fails on all metrics for all tested sequence lengths. This shows: the restriction to two token blocks in the main paper is a design choice but not a limitation.
>
> | Method | Metric | L=20 | L=30 | L=40 |
> |--|--|--|--|--|
> | Baseline | Acc | 0 | 0 | 0 |
> | | conditionalAcc | 100 | 100 | 89 |
> | | KL | 25.3 | 25.6 | 25.4 |
> | VMD | Acc | 90.6 | 87.9 | 4.82 |
> | | conditionalAcc | 100 | 100 | 100 |
> | | KL | 0.282 | 0.361 | 22.8 |
>
> ***QB2: In Alg. 2, input is a fully or partially masked sequence $x$. But $z$ is sampled from a Gaussian. When a partially masked input sequence is available, is it used to obtain $z$? If not, why?***
>
> > Our goal is to demonstrate feasibility of VMD, so we adopt the simplest setting: latent does not depend on noised data $x_t$. It is indeed possible to use $x_t$ to obtain $z$ by introducing a prior $p_\psi(z \mid x_t)$. This creates a KL term of the form $D_\text{KL}(q_{\phi}(z \mid x_0, x_t) \| p_\psi(z \mid x_t))$ which involves two moving distributions. This coupling makes optimization more involved. We tested this on synthetic data and observed no significant improvement (see below). When scaled to Sudoku data, training became unstable. When fixing the variance of $p_\psi$ to stabilize training, results were slightly worse. Details are shown in Appendix E.3. We view this noised data dependent prior as a great direction for future work.
>
> D1 ContPairs4
>
> | KL | B=2 NFE=4 | B=2 NFE=2 | B=1 NFE=4 | B=1 NFE=1 |
> |--|--|--|--|--|
> | Baseline | 0.007±0.000 | 2.298±0.189 | 0.010±0.000 | 9.422±0.253 |
> | $p(z)$ | 0.012±0.008 | 0.045±0.007 | 0.029±0.004 | 0.099±0.007 |
> | $p_\psi(z\|x_t)$ | 0.011±0.005 | 0.036±0.003 | 0.004±0.000 | 0.056±0.002 |
>
> | Accuracy | B=2 NFE=4 | B=2 NFE=2 | B=1 NFE=4 | B=1 NFE=1 |
> |--|--|--|--|--|
> | Baseline | 99.4±0.04 | 10.1±0.03 | 99.3±0.02 | 0.1±0.00 |
> | $p(z)$ | 100±0.00 | 97.8±0.01 | 100±0.00 | 93.3±0.01 |
> | $p_\psi(z\|x_t)$ | 100±0.00 | 97.6±0.02 | 100±0.00 | 94.7±0.00 |
>
> D2 RandPairs4
>
> | KL | B=2 NFE=4 | B=2 NFE=2 | B=1 NFE=4 | B=1 NFE=1 |
> |--|--|--|--|--|
> | Baseline | 0.018±0.001 | 7.656±0.138 | 0.015±0.000 | 7.493±0.211 |
> | $p(z)$ | 0.017±0.011 | 0.053±0.004 | 0.475±0.007 | 0.093±0.003 |
> | $p_\psi(z\|x_t)$ | 0.031±0.002 | 0.063±0.004 | 0.015±0.002 | 0.108±0.004 |
>
> | Accuracy | B=2 NFE=4 | B=2 NFE=2 | B=1 NFE=4 | B=1 NFE=1 |
> |--|--|--|--|--|
> | Baseline | 99.3±0.04 | 1.1±0.01 | 99.1±0.03 | 1.1±0.01 |
> | $p(z)$ | 100±0.00 | 96.3±0.01 | 100±0.00 | 94.1±0.00 |
> | $p_\psi(z\|x_t)$ | 100±0.00 | 96.9±0.02 | 100±0.00 | 92.6±0.02 |
>
> ***QB3: Does sampling different $z$ for the same starting sequence result in different outputs?***
>
> > Yes, sampling different $z$ produces different outputs for the same starting sequence.
>
> ***QB4: "NFE" used in Table 3 is defined in the appendix, but not in the main text.***
>
> > We now define NFE in Sec. 4.1.

---

> > ### Comment · Reviewer_yBdK · 2025-11-26
> > **Response from reviewer yBdK**
> >
> > The authors' rebuttal has addressed my concerns adequately. I have no further concerns.
> >
> > Also, in the authors' response to other reviewers, I saw that the architecture was modified to include an adaptive normalization layer. I could not find its mention in the revised PDF. Also, why did a normalization layer improve VMD's performance when its core idea has not changed?

---

> > > ### Author Response · Authors · 2025-11-28
> > >
> > > ***QB5: Also, in the authors' response to other reviewers, I saw that the architecture was modified to include an adaptive normalization layer. I could not find its mention in the revised PDF. Also, why did a normalization layer improve VMD's performance when its core idea has not changed?***
> > >
> > > > Thanks a lot for your time. The revised PDF describes the new conditioning scheme in Section 4.3 and Appendix C.1 and C.2 (blue highlighted part). To make the new conditioning scheme as explicit as possible, we provide the exact formulation below and added it to Appendix C.1 and C.2 (blue highlighted part).
> > >
> > > > Please note that the core idea of VMD is unchanged. The latent variable models dependencies among tokens, and the training objective remains identical to the one discussed in the original submission. The improvement is due to a revised deep net architecture which provides a more flexible mechanism for incorporating latent information without increasing trainable parameters, as described next.
> > >
> > > > Instead of the simple addition $x_{\text{emb}} = x_{\text{emb}} + z_{\text{emb}}$, which we used in the original submission, we now follow the adaLN-Zero formulation in DiT [1] and use the latent as a conditioning signal.
> > >
> > > > For synthetic data, within each DiT block we first pass the latent embedding to an MLP and get 6 conditioning parameters $(\gamma_1, \beta_1, \alpha_1, \gamma_2, \beta_2, \alpha_2) = \mathrm{MLP}(z_{\text{emb}})$. Then we inject these conditioning parameters within each DiT block as $h_1 = x_{\text{emb}} + \alpha_1 \odot \mathrm{MSA} (\mathrm{LN}(x) \odot \gamma_1 + \beta_1)$ and $h_2 = h_1 + \alpha_2 \odot \mathrm{FFN} ( \mathrm{LN}(h_1) \odot \gamma_2 + \beta_2)$ where MSA is a multi-head self-attention layer, LN is a layer norm, and FFN is a point-wise feed forward network.
> > >
> > > > For the Sudoku data, the 6 conditioning parameters $(\gamma_1, \beta_1, \alpha_1, \gamma_2, \beta_2, \alpha_2)$ are shared across all DiT blocks. All other computations ($h_1$, $h_2$) remain identical to the ones used for synthetic data. This reduces the number of learnable parameters while preserving the expressive conditioning effect of the latent variable.
> > >
> > > > Please reach out with any additional questions that you may have.
> > >
> > > > References:
> > > [1] Peebles, et al. Scalable Diffusion Models with Transformers. ICCV, 2023.

---

### Official Review · Reviewer_6okp · 2025-11-02

**Soundness:** 2
**Presentation:** 3
**Contribution:** 1
**Rating:** 2
**Confidence:** 4

**Summary:**

The token independence problem in masked diffusion can lead to degraded generation quality, which has been observed by several previous works. This paper presents a latent variable method, called VMD, for modeling the dependencies among tokens in masked diffusion. The authors also extend VMD to a block diffusion and remasking scheme. The effectiveness of VMD is validated on the artificial data sets, which can also be deemed as a contribution to experimental design.

**Strengths:**

- The presentation of this paper is clear, and it's easy for the readers to understand the whole methodology.
- The extension to the block diffusion and remasking scheme is straightforward but meaningful.
- The experimental design in Sections 4.1 and 4.2 can clearly demonstrate the VMD's ability to modeling dependencies, and can also be viewed as a contribution.

**Weaknesses:**

- The methodology of VMD is almost identical to the VADD model (https://arxiv.org/abs/2505.17384). As far as I know, VADD is the first work to consider using a latent variable model to define the transition probability $p_\theta(x_0|x_t)$, using a VAE framework for training, and discussing the related sampling framework. Specifically,
       (a) model definition. Equation (3) in VMD is similar to equation (6) in VADD,
       (b) training objective. Equation (5) in VMD is similar to equation (9) in VADD.
       (c) sampler. Alg 2 in VMD is similar to Alg 2 in VADD. The authors should cite the VADD paper and clearly state their unique contribution.

- The sampling algorithm (Alg 2) of VMD assumes a fixed latent variable $z$ and argmax sampling. This is different from the latent variable model definition in equation (3). The authors should explain the rationale behind their design.

- In line 218, different blocks employ independent latent variable priors. This is not very intuitive for me, as different blocks should be correlated with each other. Could the authors please explain why they consider independent priors for blocks?

- The experiments are limited. The authors only test the text generation quality on the text8 data set. However, baseline methods, including MDLM, consider at least an OpenWebText data set with at least a GPT-2 model size. Experiments on larger backbone and data sets would better demonstrate VMD's superiority.

**Questions:**

Please see the weaknesses.

---

> ### Author Response · Authors · 2025-11-22
> **Response to Reviewer 6okp**
>
> Thanks for your time and for recognizing the clarity of our presentation and the value of the block diffusion and remasking extensions.
>
> ***QA1: Methodology almost identical to VADD. The authors should cite the VADD paper and clearly state their unique contribution.***
>
> > Thanks for pointing out VADD. We apologize for not having been aware of it. VMD and VADD were developed concurrently and independently. In the revised paper, we cited and discussed VADD in the related work: Although both methods use a similar training objective, VMD introduces a variational latent variable together with a block structure. This results in more control over the range and structure of the latent representation. Our experiments, including synthetic tasks that explicitly control token dependence, are designed to demonstrate these effects. This isn’t discussed in VADD.
>
> ***QA2: Alg. 2 assumes a fixed latent variable $z$ and argmax sampling. This differs from the latent variable model in Eq. (4). Explain.***
>
> > Note, in our formulations (Eqs. (4) and (6)), the latent variable $z$ is defined to be time-independent. Alg. 2 is consistent with this definition. It is also possible to resample $z$ at every step. We include an ablation study in Appendix E.1, showing that the two choices lead to similar results. For simplicity, we adopt a fixed z during sampling. We added this explanation and the following experimental results in the revision. More details and results are shown in Appendix E.1.
>
> D1 ContPairs4
>
> | KL | B=2 NFE=4 | B=2 NFE=2 | B=1 NFE=4 | B=1 NFE=1 |
> |--|--|--|--|--|
> | Baseline | 0.007±0.000 | 2.298±0.189 | 0.010±0.000 | 9.422±0.253 |
> | Fixed $z$ | 0.012±0.008 | 0.045±0.007 | 0.029±0.004 | 0.099±0.007 |
> | Resampled $z$ | 0.008±0.000 | 0.061±0.005 | 0.030±0.003 | 0.089±0.006 |
>
> | Accuracy | B=2 NFE=4 | B=2 NFE=2 | B=1 NFE=4 | B=1 NFE=1 |
> |--|--|--|--|--|
> | Baseline | 99.4±0.04 | 10.1±0.03 | 99.3±0.02 | 0.1±0.00 |
> | Fixed $z$ | 100±0.00 | 97.8±0.01 | 100±0.00 | 93.3±0.01 |
> | Resampled $z$ | 100±0.00 | 98.5±0.05 | 100±0.00 | 92.8±0.04 |
>
> ***QA3: Different blocks employ independent latent variable priors. Shouldn’t different blocks be correlated with each other? Explain why independent priors for blocks?***
>
> > Great question. In this work our goal is to study feasibility of the proposed model. We hence adopt the simplest setting: time-independent and block-independent latent variables. It is however possible to use block-dependent latent variables by introducing a prior model $p_\psi(z^b \mid z^{<b})$. This results in a KL term of the form $D_\text{KL}(q_{\phi}(z^{b} \mid x_0^{\leq b}, x_t^b) \|\| p_\psi(z^b \mid z^{<b}))$ which involves two moving distributions. This coupling might make optimization less stable in practice. We tested this variant on synthetic data and observed a small improvement. But when scaling to text8, training became unstable and the final performance was worse than the block independent version. Details are provided in the tables below and in the newly added Appendix E.2. For clarity and stability, we use the independent prior in the main paper. We view block dependent priors as a great direction for future work.
>
> D1 ContPairs4
>
> | KL | B=2 NFE=4 | B=2 NFE=2 | B=1 NFE=4 | B=1 NFE=1 |
> |--|--|--|--|--|
> | Baseline | 0.007±0.000 | 2.298±0.189 | 0.010±0.000 | 9.422±0.253 |
> | Block independent $z$ | 0.012±0.008 | 0.045±0.007 | 0.029±0.004 | 0.099±0.007 |
> | Block dependent $z$ | 0.010±0.003 | 0.034±0.001 | 0.013±0.002 | 0.071±0.008 |
>
> | Accuracy | B=2 NFE=4 | B=2 NFE=2 | B=1 NFE=4 | B=1 NFE=1 |
> |--|--|--|--|--|
> | Baseline | 99.4±0.04 | 10.1±0.03 | 99.3±0.02 | 0.1±0.00 |
> | Block independent $z$ | 100±0.00 | 97.8±0.01 | 100±0.00 | 93.3±0.01 |
> | Block dependent $z$ | 100±0.00 | 97.7±0.03 | 100±0.00 | 94.7±0.02 |
>
> ***QA4: Authors only test text generation on text8. Baselines consider at least OpenWebText data with at least a GPT-2 model. Experiments on larger backbone and data would better demonstrate VMD's superiority.***
>
> > Thank you for the suggestion. Due to GPU limitations we were not able to run large scale text experiments in the initial submission. We are currently running VMD on a DiT backbone with a size comparable to GPT-2 small and expect to obtain results within the coming week. We will update the paper once these experiments finish.

---

### Author Response · Authors · 2025-11-22

We appreciate the reviewers’ comments and suggestions. We have revised the paper based on their feedback and respond to each reviewer below.

In the revised manuscript, we updated the experimental results and included additional ablation studies. First, we updated the model architecture to strengthen how the latent variable is incorporated into the decoder. This change keeps the overall model size unchanged but significantly improves performance. Based on this updated architecture, we reran all experiments on synthetic data and Sudoku, and the improved results are shown in the revision.

We have also added a comprehensive set of ablation studies to address questions about the design of the latent variable. Appendix E now includes ablations on time dependent latents $z_t$, block dependent latents $p_\psi(z^b \mid z^{<b})$, noised data dependent latents $p_\psi(z \mid x_t)$, and updating the latent during sampling using the encoder distribution $q_\phi(z \mid x_0, x_t)$. These studies provide a more complete view of how different latent formulations affect stability and performance.

Together, these revisions directly address the concerns raised by the reviewers and we hope they clarify and strengthen the contribution of our work. We are also running experiments on a larger DiT backbone to provide a more comprehensive text evaluation and will update the paper once these results are ready.

---

### Author Response · Authors · 2025-12-03
**Final Remarks**

We thank all reviewers for their thoughtful feedback. Reviewers described the paper as “well written and easy to follow, with clean mathematical derivations,” “clear and easy to understand,” and “presented with a nice exposition” (6okp, dRUz, yBdK). Reviewers also highlighted the conceptual contributions, noting that the work studies an “interesting problem,” that the “idea is novel,” and that the formulation is “cleanly stated” with an “ELBO objective that is theoretically grounded” (yBdK, MC7Q, GQc8, 8mJQ). Several reviewers emphasized the importance and potential impact of the problem, calling it a “key problem” and one that “can impact real-world large language models” (yBdK, MC7Q). The experimental design was repeatedly recognized as a strength, with reviewers noting that the experiments are “well designed,” “comprehensive,” “diverse,” and that they “demonstrate the method’s ability to model dependencies” and “show its efficacy on synthetic datasets in a convincing manner” (6okp, 8mJQ, dRUz, GQc8, MC7Q).

We would also like to clarify the scope of our work. Our primary goal is to demonstrate that variational masked diffusion is a feasible and effective framework for modeling token dependencies. The suggestions from reviewers on richer priors and alternative sampling schemes are valuable future directions. To answer these questions and to show that our framework is flexible, we added a comprehensive ablation study in Appendix E. For the prior model, we explored block dependent latents $p_\psi(z^b \mid z^{<b})$ (Appendix E.2) and noised data dependent latents $p_\psi(z \mid x_t)$ (Appendix E.3). For sampling, we examined generating a new latent $z_t$ at each step (Appendix E.1) and updating the latent during sampling with the encoder $q_\phi(z \mid x_0, x_t)$ (Appendix E.4). These results support the feasibility and robustness of our method while leaving more elaborate design choices as promising future directions.

Several reviewers asked whether improvements on Sudoku and text data are marginal. To answer, we studied another model architecture to more effectively incorporate latent information by replacing simple addition with an adaptive layer normalization module. Detailed updates are provided in Appendix C, with modifications highlighted in blue.

For Sudoku data, the updated results are shown below.

| Top prob (Accuracy ↑) | NFE=5 | NFE=10 | NFE=20 |
|--|--|--|--|
| MDM | 10.6%  | 14.7%  | 20.4% |
| **VMD** | **67.7%**  | **76.4%** | **80.9%** |

| Top prob margin (Accuracy ↑) | NFE=5 | NFE=10 | NFE=20 |
|--|--|--|--|
| MDM | 36.2% | 78.4%  | 91.1% |
| **VMD** | **96.9%** | **99.0%** | **99.7%** |

For text data, we trained VMD on the larger LM1B dataset. Generative perplexity and entropy are evaluated by a pretrained GPT-2 Large model using 256 samples. The new results are as follows.

| Model | Test PPL ↓ | Generative PPL ↓ | Entropy ↑ |
|--|--|--|--|
| BD3-LM Block size 4 | 46.26 | 123.243 ± 31.669 | 4.149 ± 0.415 |
| **VMD Block size 4**  | **44.88** | **107.430 ± 12.600** | **4.281 ± 0.076** |

Via the revision, we answered all reviewers’ questions and corroborated statements with empirical results. We clarified the scope of the work and demonstrated that variational masked diffusion (VMD) is a flexible and principled framework for modeling token dependencies. The improved Sudoku and LM1B results confirm that VMD provides meaningful gains beyond marginal improvements. We hope that the clarified contributions and strengthened results help highlight the value of VMD as a promising direction for future generative modeling research.

---

### Meta-Review · Area_Chair_fTk6 · 2026-01-05

**Summary:**

This paper proposes a variational based method to enable more efficient inference of Masked Diffusion Models. In standard MDMs, the dependency among tokens are not captured in the model. As a result, multi-token unmasking at one step will introduce bias. This limits the inference efficiency of MDMs. The authors propose to utilize a latent variable as in VAE to bring this dependence back. The authors further present a block diffusion version of the method based on recent developments of block diffusion LLMs.

**Reviewer Concerns:**

There are three major criticisms. First, one important reference titled “Variational Autoencoding Discrete Diffusion with Enhanced Dimensional Correlations Modeling” is overlooked. The novelty with respect to this reference is incremental at best. Second, in the text related experiments, the improvement of VMD over MDM is marginal. Third, the model size used in this paper is way too small compared to SOTA, making it difficult to tell whether the proposed method will have practical impacts.

**Reviewer Scores:**

I don't see the possibility of increasing score

---

### Decision · Program_Chairs · 2026-01-26

Reject